

**Implementation of salt-induced freezing point depression function into**
**CoupModel_v5 for improvement of modelling seasonally frozen soils**
Mousong Wu[1,2,3], Per-Erik Jansson[2], Jingwei Wu[1*], Xiao Tan[1,4], Kang Wang[1], Peng Chen[5], Jiesheng
Huang[1]
*1.    State Key Laboratory of Water Resources and Hydropower Engineering Science, Wuhan University,*
*430072 Wuhan, Hubei, China*
*2.    Department of Sustainable Development, Environmental Science and Engineering, KTH Royal*
*Institute of Technology, 10044 Stockholm, Sweden*
*3.    Department of Physical Geography and Ecosystem Science, Lund University, 23362 Lund, Sweden*
*4.    State Key Laboratory of Hydraulics and Mountain River Engineering, College of Water Resource*
*& Hydropower, Sichuan University, 610065 Chengdu, Sichuan, China*
*5.    Department of Geology, Lund University, 23362 Lund, Sweden*
**Abstract**
Soil freezing/thawing is important for soil hydrology and water management in cold regions. Salt in
agricultural field impacts soil freezing/thawing characteristics and therefore soil hydrologic process. In
this context, we conducted field experiments on soil water, heat and salt dynamics in two seasonally
frozen agricultural regions of northern China to understand influences of salt on cold regions hydrology.
We developed CoupModel by implementing impacts of salt on freezing point depression. We employed a
Monte-Carlo sampling method to calibrate the new model with field observations. The new model
improved soil temperature mean error (ME) by 16% to 77% when new freezing point equations were
implemented into CoupModel. Nevertheless, we found that parameters related to energy balance and soil
freezing characteristics in the new model were sensitive to soil heat and water transport at both sites.

*Correspondence author.
Email: jingwei.wu@whu.edu.cn



However, a systematic model sensitivity and calibration has shown to be able to improve model performance, with mean values of $R^2$ from behavioral simulations for soil temperature at 5 cm depth as high as 0.87 and 0.90, and mean value of $R^2$ for simulated soil water (liquid or total water contents at 5 cm depth) of 0.31 and 0.80 at site Qianguo and site Yonglian, respectively. This study provided a new approach considering influences of salt on soil freezing/thawing in numerical models and highlighted the importance of salt in soil hydrology of seasonally frozen agricultural soils.

Keywords: Saline soil; freezing point; seasonal frost; sensitivity; soil hydrology

## 1. Introduction

Soil freezing and thawing processes have long been recognized for its importance in not only engineering applications (e.g., construction of roads and pipelines) (Jones, 1981; Hansson et al., 2004; Wettlaufer and Worster, 2006), but also environmental issues (e.g., soil erosion, flooding, and pollutants migration) (Andersland et al., 1996; Seyfried and Murdock, 1997; Baker and Spaans, 1997 ; McCauley et al., 2002). Knowledge on soil freezing and thawing could uncover mechanisms on water and salt distribution in soil (Baker and Osterkamp, 1989), on frost heaving (Wettlaufer and Worster, 2006), on waste disposal technology (McCauley et al., 2002), as well as on climate change and water management in cold regions (Lopez et al., 2007).

Laboratory and field experiments on hydrological characteristics of freezing/thawing soils have been conducted to understand soil hydrology in cold regions. Most of the experiments focused on soil freezing characteristics under various climate and soil conditions (Williams, 1964; Black and Tice, 1989; Spaans and Baker, 1996; Azmatch et al., 2012), regional water and energy balance in winter (Fuchs et al., 1978; Baker and Spaans, 1997; Hayashi et al., 2004; Watanabe et al., 2013; Zhou et al., 2014). There were very few studies on salt transport in frozen soils, except for frost heaving. Cary et al. (1979) found salt can decrease frost heaving and increase infiltration in frozen soils based on observations. Konrad and McCammon (1990) found the expulsion of salt from ice is dependent on freezing rate of soil.



Hydrological effects of salt in cold regions have not been deeply explored. Wang et al. (2016) compared
water and salt fluxes in two agricultural fields same as in this study, and detected different flow
characteristics of salt during soil freezing and thawing seasons. They demonstrated that salt expulsion and
dispersion are not negligible in frozen soils. Wu et al. (2016a) found that evaporation during winter was
controlled by soil salt and groundwater in field frost tube experiments, in Inner Mongolia, China. They
also demonstrated that water, heat and salt transport in frozen soils were coupled, and due to spatial
heterogeneity of soil properties and technical difficulties in soil freezing/thawing experiments,
measurements contained large uncertainties.
Numerical models on soil freezing/thawing have been put forward by many. Jansson and Karlberg
(2004) developed a coupled process-based model—CoupModel, to simulate water, heat as well as salt
transport in frozen soil. CoupModel is a process-based model with detailed descriptions on coupled water
and heat transport in frozen and unfrozen soils. It has shown to be one of the most robust models among
other models taking soil freezing/thawing into account (e.g., SWAP, DRAINMOD, SWAT, HBV, VIC,
and ATS etc). This model was developed and applied to forests (Gustafsson et al., 2004; Wu et al., 2013),
agricultural field (Wu et al., 2011), permafrost (Zhang et al., 2012; Scherler et al., 2013) and other
ecosystems (Okkonen and Kløve, 2011; Khoshkhoo et al., 2015).
However, there were large uncertainties in modeling soil freezing and thawing due to the complexity
of phase change and coupled processes. To reduce uncertainties in modeling, uncertainty analysis method
was always introduced by combining experimental data with numerical models in calibration of the
models for better representativeness of reality. The generalized likelihood uncertainty estimation (GLUE)
technique (Beven and Binley, 1992) is the commonly used method for uncertainty analysis in
environmental modeling. Instead of searching for an optimal parameter set, the GLUE method generates
ensembles of parameter sets that show equally good performance in simulations, called 'equifinality' by
Beven (2006). GLUE was performed by randomly sampling the parameter space within their ranges using


Monte-Carlo sampling method, and then by selecting behavioral simulations using criteria applied to
performance metrics.

In this study, we performed experiments on water, heat and salt transport at two seasonal frost sites

located in northern part of China. They are different in climate and in soil conditions, but are both
important agricultural regions in northern part of China. The Hetao Irrigation District in China is a typical
arid agricultural region suffering from soil salinization due to saline water irrigation, extensive
evaporation, as well as soil freezing/thawing (Li et al., 2012). The Songyuan Irrigation District is a typical
paddy rice grown region in northeastern part of China, suffering from high salinity due to over-
development of salinized field into agricultural field (Liu et al., 2001). Soils in both regions go through
freezing/thawing during winter and suffer from salinization in spring. These two sites are crucial in water
resources management of China under the concept of water-saving agriculture. Wu et al. (2016b)
performed calibration on soil water and heat transport based on one plot in the experimental field in Hetao
Irrigation District in Inner Mongolia and found that the influences of salt on soil freezing should be taken
into account. Wang et al. (2016) conducted field experiments and analyzed water and solutes transport
characteristics at these two above-mentioned sites and demonstrated that salt transport in frozen soils is
more complicated than in unfrozen soils due to diffusion and solute rejection. Thus, we developed
CoupModel by considering impacts of salt on freezing, and applied the new model to the agricultural sites
for modeling water, heat and salt in two seasonal frost soils. The main objective was to 1) develop
CoupModel by considering effects of salt on freezing point; 2) identify sensitivity of parameters; 3)
analyze uncertainty in modeling soil hydrology in seasonal frost agricultural soils.
**2.   Material and Methods**
*2.1 Study sites*

Experiments were conducted at two agricultural sites of northern China. One site is located in Qianguo

Irrigation District of Songyuan, Jilin province, China (lat: 45.24°, lon: 124.60°, hereafter referred as site


Qianguo) (**Fig. 1**). Field experiment at site Qianguo was conducted during 2011/2012 winter. Annual
precipitation at site Qianguo is 451 mm and annual mean air temperature is 5.1 $^{\circ}$C (averaged from 2011 to
2012). This study site is typical for its soil texture classified as clay, which has a high bulk density, low
porosity, and low hydraulic conductivity (**Table 1**). Soil profile at site Qianguo is homogeneous, with
porosity of 0.46 and bulk density of 1.42 g cm$^{-3}$. The water table in this area fluctuates between 1.5 and
2.0 m. Maximum frost depth at site Qianguo is 1.2 m. Six plots (2×2 m$^2$ for each) were selected in a
paddy field, which was cultivated with paddy rice from May to October. On 2011/10/09, 20 mm NaBr
solution containing 6.5 g L$^{-1}$ Br$^{-}$ was applied to each plot to from the initial profile for Br$^{-}$. Before
spraying the solution, stubbles were removed from the plots and surface was ploughed to depth of 20 cm.

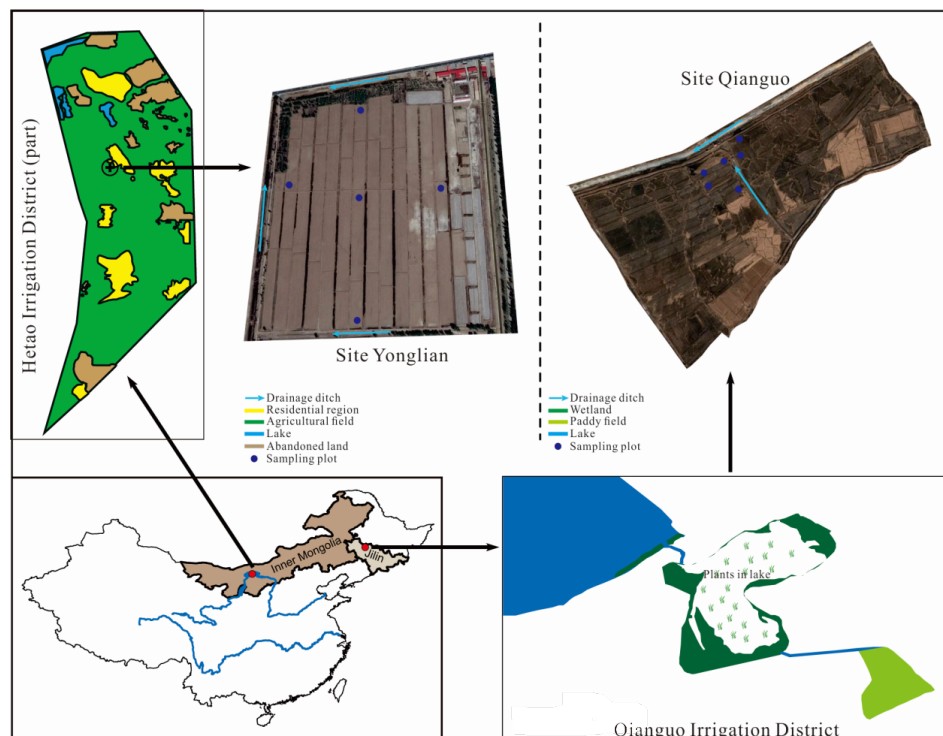

**Fig. 1** Locations of the study sites. Site Yonlian is located at the middle part of Hetao Irrigation District of
Inner Mongolia Autonomous Region, northern part of China, site Qianguo is located at Songyuan County
of Jilin Province, northeastern part of China.





The other site is located at Yonglian experimental station in Hetao Irrigation District of Inner
Mongolia Autonomous Region, China (lat: 41.13$^{o}$, lon: 108.00$^{o}$, hereafter referred as site Yonglian) (**Fig.**
**1**). Field experiment at Yonglian was conducted during 2012/2013 winter from October 1$^{st}$ to April 30$^{th}$.
Annual precipitation at this site is 140 mm, and annual mean air temperature is 6.4 $^{o}$C (averaged for
2012~2013). Soil profile is heterogeneous, with porosity of 0.42-0.46 and bulk density of 1.44-1.53 g cm$^{-}$
$^{3}$ (averaged over 5 plots). Water table fluctuates between 1.5 and 3 m during winter. From November 4$^{th}$
to 6$^{th}$ of 2012, flooding irrigation (autumn irrigation) with 250 mm water was applied to the field for
leaching salt that accumulated during growing season. Soil profile mean salt content (mainly NaCl) is 0.1%
g g$^{-1}$ for the study site, and irrigation water electrical conductivity is 0.5 mS cm$^{-1}$. Before autumn
irrigation, five plots (2×2 m$^{2}$) were selected for experiment at different parts of the agricultural field, and
ploughed to 20 cm depth.
**Table 1** Soil physical and chemical properties at two study sites Qianguo and Yonglian.

| Site | Depth (cm) | Clay (%) | Silt (%) | Sand (%) | Organic matter (%) | Bulk density (g/cm$^3$) | Porosity (-) |
|------|-----------|----------|----------|----------|--------------------|-----------------------|--------------|
| NE | 0-140 | 32.40 | 38.40 | 29.30 | 1.76 | 1.42 | 0.46 |
| IM | 0-10 | 27.91 | 17.20 | 54.89 | 0.53 | 1.49 | 0.44 |
| | 10-20 | 37.46 | 21.65 | 40.90 | 1.08 | 1.45 | 0.45 |
| | 20-30 | 31.69 | 48.39 | 19.92 | 0.77 | 1.44 | 0.46 |
| | 30-40 | 34.08 | 34.32 | 31.61 | 0.73 | 1.45 | 0.45 |
| | 40-60 | 28.74 | 34.85 | 36.40 | 0.53 | 1.47 | 0.45 |
| | 60-80 | 34.67 | 30.45 | 34.88 | 0.77 | 1.45 | 0.45 |
| | 80-100 | 17.65 | 18.46 | 63.88 | 1.14 | 1.53 | 0.42 |
| | 100-140 | 14.81 | 35.05 | 50.15 | 0.44 | 1.53 | 0.42 |


*2.2 Experimental design*



TDR probes (Model: CS605, Campell Scientific Inc.) were installed at site Qianguo to detect liquid
water content. Due to difficulty in long-term maintaining of TDR system in rural regions, only daily
liquid water was manually recorded with a datalogger (TDR 100; Campbell Scientific Inc.). TDR probes
were calibrated in laboratory with unfrozen soil, and the precision of calibration was maintained with $R^2$
of 0.97. TDR probes were then inserted horizontally into the soil pit (10 m apart from the experimental
plots) from 5 cm to 100 cm depth with 10 cm interval. PT100 temperature sensors were installed at the
same depth as TDR probes, and the daily temperature data were collected.
During soil freezing/thawing season at site Qianguo, 7 sampling dates were chosen (2011/10/09,
2011/11/09, 2011/11/25, 2011/12/20, 2012/02/15, 2012/04/10, 2012/04/20), and soil samples from 0 to
100 cm with 10 cm interval were collected for determining total water content and $Br^-$ content. An electric
drill (5 cm in diameter, 10 cm in length) was used for sampling frozen soil for every 10 cm depth. Total
water content was determined by oven-dry method. $Br^-$ content was determined by diluting 50 g wet soil
into 250 mL deionized water, and measuring the electrical potential (mV) using an electrical potential
meter (MP523-06). Then the electrical potential was converted into $Br^-$ concentration by a pre-calibrated
relationship between $Br^-$ concentration and electrical potential (calibration $R^2$=0.99). Soil temperature and
liquid water content at 4 depths (5, 15, 25, and 35 cm) from site Qianguo were used to calibrate
CoupModel, while soil water storage and salt storage at various depths (0-10 cm, 0-40 cm, 0-100 cm)
estimated form soil profile samplings were used to validate the model.
Total water content and $Cl^-$ content from 0 to 100 cm with 10 cm interval at site Yonglian were
sampled at 14 dates from October 2012 to April 2013 (2012/10/16, 2012/10/27, 2012/11/10, 2012/12/04,

141     2012/12/15, 2012/12/26, 2013/01/05, 2013/01/14, 2013/01/25, 2013/03/05, 2013/03/14, 2013/03/25,

2013/04/07, 2013/04/18). The sampling and measurement methods for total water content and $Cl^-$ content
were the same as those at site Qianguo. During experimental period (2012/10/01-2013/04/30), hourly soil
temperatures at 5, 15, 25 and 35 cm depth were recorded by the PT100 temperature sensors from the
micro-meteorological station in the field. Groundwater table depth was measured every day during the



autumn irrigation and drainage period (2012/11/4 to 2012/11/15), and for every five days during the rest
time of the winter. Soil started freezing from November 12[th], 2012, and total thawed on April 30[th], 2013.
Measuring of groundwater table depth was conducted manually by putting a roped copper cup into
observation well, and then measuring the rope length when the cup touched water (by hearing the voice).
Soil temperature, soil total water content at 4 depths (5, 15, 25 and 35 cm) and groundwater table depth
were used to calibrate CoupModel, while water storage and salt storage at different depths (0-10 cm, 0-40
cm, 0-100 cm) estimated from soil profile samplings were used to validate the model.
Meteorological data e.g. air temperature, humidity, radiation, wind speed, and precipitation, were
obtained from the nearest meteorological station at each site with hourly-resolution from October 1[st],
2011 to April 30[th], 2012  and from October 1[st], 2012 to April 30[th], 2013 at site Qianguo and site Yonglian,
respectively.
**3.  CoupModel_v5**
Model domain covered from soil surface to 6 m depth, with unit area considered. Soil profile was
discretized into 16 layers, with 10 cm thickness each layer from 0 to 40 cm, 20 cm thickness from 40 cm
to 2 m, and 1 m thickness from 2 m to 6 m. Input meteorological data were hourly, and model time step
was set as hourly. Numerical solution of water, heat and salt transport in soils was based on forward
difference method. Model performance metrics on different output variables was calculated automatically
using modules implemented into CoupModel_v5. Major model processes considered in this study were
described in the following sections.
*3.1  Soil water processes*
CoupModel solved coupled differential equations for water and heat transfer (Jansson, 2012). Water
flow in the soil matrix was described by Richards equation:
$$\frac{\partial \theta}{\partial t} = \frac{\partial}{\partial z}\left[k_w\left(\frac{\partial \psi}{\partial z}-1\right)\right]+\frac{\partial}{\partial z}\left(D_v\frac{\partial C_v}{\partial z}\right)-\frac{\partial q_{bypass}}{\partial z} \tag{1}$$


where $\theta$ is water content (m$^3$ m$^{-3}$); $k_w$ is hydraulic conductivity (m s$^{-1}$); $\psi$ is matric potential (m); $D_v$ is
vapor diffusion coefficient (m$^2$ s$^{-1}$); $C_v$ is vapor density (m$^3$ m$^{-3}$); $q_{bypass}$ is the bypass flow in macro
pores (m s$^{-1}$); $z$ is depth to soil surface (positive downward) (m); and $t$ is time (s).
Vapor flow (second term inside brackets on right side of **Equation (1)**) in soil was determined by
vapor gradient between two layers and diffusion coefficient, adjusted by tortuosity $d_{vapb}$ (**Equation (2)**).

$$q_v = d_{vapb} D_0 f_a \frac{\partial C_v}{\partial z} \tag{2}$$

where $d_{vapb}$ is a parameter accounting for tortuosity; $D_0$ is the diffusion coefficient for free air (m$^2$ s$^{-1}$);
$f_a$ is the soil air content (m$^3$ m$^{-3}$); $C_v$ is vapor density (m$^3$ m$^{-3}$); and $D_v$ is vapor diffusion coefficient (m$^2$ s$^{-
1}$).
The diffusion coefficient for free air, $D_0$ was a function of soil temperature (**Equation (A1)** in **Table**
**A2**), and vapor density $C_v$ was calculated from vapor pressure (**Equation (A1)**), which was estimated
from soil matric potential and soil temperature (**Equation (A1)**).
Infiltration through frozen soil was estimated separately for the low- and high-flow domains (Stähli et
al., 1996). CoupModel took the preferential flow in macropores into account using a bypass routine when
excess water entering the soil was routed directly to the next underlying soil layer through the high-flow
domain (Jansson, 2012). Infiltration into soil was determined by the soil adsorption rate adjusted by a soil
matric water adsorption coefficient $a_{scale}$ (**Equation (5)**). When infiltration water was larger than soil
adsorption rate, bypass flow would occur. Bypass flow in macro pores was determined by

$$q_{bypass} = \begin{cases} 0 & 1 < q_{in} < s_{mat} \\ q_{in} - q_{mat} & q_{in} \geq s_{mat} \end{cases} \tag{3}$$

$$q_{mat} = \begin{cases} \max\left(k_w(\theta)\left(\frac{\partial \psi}{\partial z} + 1\right), q_{in}\right) & 1 < q_{in} < s_{mat} \\ s_{mat} & q_{in} \geq s_{mat} \end{cases} \tag{4}$$





$$s_{mat} = a_{scale}a_r k_{mat} pF \qquad (5)$$

where $s_{mat}$ is soil adsorption rate (m s$^{-1}$); $a_{scale}$ is soil matric water adsorption coefficient, $a_r$ is a
geometry coefficient to describe thickness ratio to horizontal scale of each soil layer; $k_{mat}$ is matric
maximum hydraulic conductivity (m s$^{-1}$); and $pF$ is pF value of soil.
Flow in low-flow domain obeyed Darcy's law and soil water retention curve was determined by
Brooks and Corey (1964) equation (**Equation (A3)**), with the air entry at different layers to be adjusted in
calibration. Hydraulic conductivity in low-flow domain was calculated from the Mualem (1976) equation
(**Equation (A4)**). In frozen soil, hydraulic conductivity was modified for high-flow (Stähli et al., 1996).
In high-flow domain, water flow was modeled by gravitational flow under unit gradient, and hydraulic
conductivity was adjusted by using impedance factor $c_{\theta,i}$ in high-flow domain (**Equation (6)**):
$$k_{fh} = e^{-\frac{\theta_i}{c_{\theta,i}}} \left( k_w\left(\theta_{tot}\right) - k_w\left(\theta_{lf} + \theta_i\right) \right) \qquad (6)$$

where $k_w(\theta_{tot})$ is hydraulic conductivity for pores saturated with water (m s$^{-1}$); $k_w\left(\theta_{lf} + \theta_i\right)$ is hydraulic
conductivity when water flow in low domain with ice existence (m s$^{-1}$); $\theta_i / c_{\theta,i}$ is reduced factor; $c_{\theta,i}$ is
impedance factor; $\theta_{tot}$ is total water content in high- and low-flow domains (m$^3$ m$^{-3}$); $\theta_{lf}$ (=$d_1\theta_{wilt}$,
**Equation (18)**) and $\theta_i$ ($\frac{E-H}{\Delta z L_f \rho_{ice}}$, $E$ is soil heat, J, $H$ is sensible heat, J, $L_f$ is latent heat, J kg$^{-1}$, $\Delta z$ is soil
thickness, m, $\rho_{ice}$ is ice density, kg m$^{-3}$) are the liquid water and ice content, respectively, in the low-flow
domain (m$^3$ m$^{-3}$).
The hydraulic conductivity changed at the freezing front under partially frozen conditions. To prevent
excessive water redistribution towards the freezing front, the hydraulic conductivity of partially frozen
layers was adjusted by considering ice content influences on water flow using a factor $c_{fi}$ (**Equation (7)**).
$$k_{wf} = 10^{-c_{fi}Q} k_w \qquad (7)$$





where $c_{fi}$ is impedance factor; and $Q$ is heat quality, as a ratio of ice content to total water content.
Meanwhile, the influence of soil temperature on soil hydraulic conductivity was considered, using a
linear increase factor $r_{A1T}$ and a minimum conductivity $k_{minuc}$ to adjust hydraulic conductivity at 20 $^{\circ}$C
(**Equation (A7)**).
Surface ponding of water may occur if the soil infiltration capacity was exceeded, otherwise the
infiltration rate was equal to precipitation and rates of snowmelt. If infiltration capacity was exceeded,
excess water will be transferred to the surface pool. The overland flow from surface pool was estimated
by the difference between surface water storage and maximum surface pool, $w_{pmax}$ (**Equation (8)**).
$$q_{\text{surf}} = a_{\text{surf}} \left( W_{\text{pool}} - w_{\text{pmax}} \right) \tag{8}$$
where $a_{\text{surf}}$ is an empirical coefficient, $W_{\text{pool}}$ is the total amount of water in the surface pool (m), and $w_{\text{pmax}}$
is the maximal amount of water stored on soil surface without causing surface runoff (m).
Drainage systems at two study sites were open drainage ditches, drainage at study sites was then
calculated by Hooghoudt equation combined with an empirical drainage equation to constitute a manual
drainage system, adjusted by initial drainage level $z_p$ and minimum drainage level, drain spacing $d_p$
(**Equation (A9)**), empirical groundwater level peak value $z_1$, and empirical groundwater flow peak value
$q_1$ (**Equation (A10)**). Meanwhile, initial groundwater level was set for calibration. Groundwater water
level was estimated by the soil saturation layer depth to surface.
*3.2 Soil heat processes*
Heat flow in soil was described by the heat transport equation, considering conduction, convection and
latent heat flow:
$$\frac{\partial (CT)}{\partial t} - L_f \rho_i \frac{\partial \theta_i}{\partial t} = \frac{\partial}{\partial z}\left( k_h \frac{\partial T}{\partial t} \right) - C_w \frac{\partial (q_w T)}{\partial z} - L_v \frac{\partial q_v}{\partial z} \tag{9}$$



where $C$ is soil (containing solid, water, and ice) heat capacity (J m$^{-3}$ °C$^{-1}$); $T$ is temperature (°C); $L_f$ is
latent heat of freezing (J kg$^{-1}$); $\rho_i$ is density of ice (kg m$^{-3}$); $\theta_i$ is ice content (m$^3$ m$^{-3}$); $k_h$ is thermal
conductivity soil (W m$^{-1}$ °C$^{-1}$); $q_w$ is water flux (m s$^{-1}$); $L_v$ is latent heat of vaporization (J kg$^{-1}$); and $q_v$
is vapor flux (m s$^{-1}$).
Upper boundary for soil heat flow was soil temperature at surface, calculated by the energy balance
scheme described in *Section 3.4*. Lower boundary for soil heat flow was controlled by soil temperature
fluctuation at 6 m depth, which was estimated using an analytical solution for soil heat conduction.
Soil thermal conductivity for both frozen and unfrozen soils was calculated from the Ballard & Arp
equation (Balland and Arp, 2005), adjusted by three empirical coefficients $\alpha$, $\beta$, and $a$ (**Equation (A11)**).
Thermal conductivity from the top frozen soil layer was then corrected by using a damping function,
adjusted by the maximum damping coefficient $C_{md}$ (**Equation (A12)**). When infiltration water passed the
high-flow domain, it would refreeze due to low soil temperature in frozen soils. Meanwhile, latent heat
released from refreezing would melt water in high-flow domain. This would lead to redistribution of
water between low-flow and high-flow domains. CoupModel considered the water redistribution and
adjusted it by a heat transfer coefficient $\alpha_h$ (**Equation (A13)**).
*3.3 Salt tracer processes*
Salt in CoupModel was simulated as a tracer migrating with water, neglecting diffusion. Salt transport
was simulated as Cl$^-$ transport in soil for estimate of salt tracer flux. Salt balance in soil is calculated as:
$$\frac{\partial c_{Cl}}{\partial t} = -\frac{\partial}{\partial z}(q_{mat} c_{Cl}) - \frac{\partial}{\partial z}\left(q_{bypass} c_{Cldep}\right) \tag{10}$$

where $c_{Cl}$ is concentration of Cl$^-$ (kg m$^{-3}$); $c_{Cldep}$ is salt deposition concentration (kg m$^{-3}$); $q_{mat}$ is water
flux (m s$^{-1}$), $q_{bypass}$ is bypass flow (m s$^{-1}$).





Soil salt concentration for each layer was then calculated by
$$c_{Cl}(z) = \frac{S_{Cl}(z)(1 - s_{adc}(z))}{\theta(z)\Delta z}$$    (11)
where $s_{Cl}$ is salt amount at each soil layer (kg m$^{-2}$); $s_{adc}$ is salt adsorption rate; $\theta$ is soil water content at
each layer (m$^3$ m$^{-3}$); $\Delta z$ is soil layer thickness (m).
Salt at surface was balanced by salt in precipitation and irrigation, as well as salt loss from surface
runoff. Lower and lateral boundaries for salt transport were salt leaching to groundwater, which was
proportional to drainage rate. Initial salt concentration $c_{Cl}$, precipitation salt concentration $c_{Cldep}$,
irrigation salt concentration $c_{Clirrig}$, as well as salt adsorption coefficient $s_{adc}$ at different depths were set
as calibration parameters. At site Qianguo, Br$^-$ transport was converted to Cl$^-$ transport in the simulation
in the validation of salt storage, Cl$^-$ storage at site Qianguo was then converted to Br$^-$ storage in
comparison with field observations of Br$^-$ storage.
*3.4  Energy balance processes*
Surface temperature and evaporation was calculated using energy balance method, with net short-wave
radiation balanced by latent heat, sensible heat and soil heat flux at surface:
$$R_s = L_v E_v + H_s + q_h$$    (12)
where $L_v E_s$ is the sum of latent heat flux (J m$^{-2}$ s$^{-1}$); $H_s$ is sensible heat flux (J m$^{-2}$ s$^{-1}$) and $q_h$ is heat flux
to the soil (J m$^{-2}$ s$^{-1}$).
Latent heat was calculated as below,
$$L_v E_s = \frac{\rho_a c_p}{\gamma}\frac{(e_{surf} - e_a)}{r_{as}}$$    (13)





where $r_{as}$ is the aerodynamic resistance (m$^{-1}$); $e_{surf}$ is the vapor pressure at the soil surface (Pa or in m
water); $e_a$ is the actual vapor pressure in the air (Pa or in m water); $\rho_a$ is the air density (kg m$^{-3}$); $c_p$ is the
heat capacity of air (J kg$^{-1}$ °C$^{-1}$); $L_v$ is the latent heat of vaporization (J kg$^{-1}$) and $\gamma$ is the psychometric
constant.
Sensible heat was calculated as
$$H_s = \rho_a c_p \frac{(T_s - T_a)}{r_{as}}$$
(14)

where $T_s$ is the soil surface temperature (°C); $T_a$ is the air temperature (°C); and $r_{as}$, $\rho_a$, $c_p$ are the same as
**Equation (13)**.
Soil surface heat flow was then calculated,
$$q_h = k_h \frac{(T_s - T_1)}{\frac{\Delta z_1}{2}} + L q_{v,s}$$
(15)

where $k_h$ is the thermal conductivity of the topsoil layer (W m$^{-1}$ °C$^{-1}$); $T_s$ is the soil surface temperature
(°C); $T_1$ is the middle of uppermost soil compartment temperature (°C); $\Delta z_1$ is the depth of the uppermost
soil compartment (m) and $L q_{v,s}$ is the latent water vapor flow from soil surface to the central point of the
uppermost soil layer (J m$^{-2}$ s$^{-1}$).
Surface temperature was then adjusted to make Equation (5) balanced by different fluxes at surface.
Soil surface vapor pressure was determined by soil surface temperature, water potential at top layer and
soil water gradient between soil surface and top layer. This was further corrected by an empirical factor,
which was adjusted by an adjustment coefficient $\psi_{eg}$, and the surface water balance (**Equation (A14)**),
which was adjusted by maximum soil surface water deficit $s_{def}$ and maximum soil surface water excess
$s_{excess}$ (**Equation (A15)**).





Aerodynamic resistance for stable atmosphere was calculated using the Richardson equation. Then the
aerodynamic resistance for stable atmosphere was adjusted by the momentum roughness length of soil
and snow surface $z_{0M}$ ($z_{0M,snow}$) (**Equation (A16)**), and the heat roughness length of surface $z_{0H}$ was
derived from $z_{0M}$ and $kB^{-1}$ (**Equation (A17)**). In addition, when surface was at extreme stability
conditions, aerodynamic resistance was then adjusted by using a windless exchange coefficient $r_{a,max}^{-1}$
(**Equation (A18)**).
Soil evaporation was adjusted by maximum soil water condensation rate $e_{max,cond}$ considering the
influences of condensation of water on evaporation (**Equation (A19)**). Net radiation was estimated by
Konzelmann equation with two formulae to calculate longwave radiation and was adjusted by an
empirical coefficient $r_{k1}$ (**Equation (A20)**). Snow melting was determined by solving energy balance
equation in snowpack using the same scheme as soil surface energy balance calculation. Snow mass
balance was then estimated based on temperature change in snowpack as well as snow age. Snow thermal
conductivity was calculated from snow density with an adjustment factor $s_k$ (**Equation (A21)**). Soil
albedo was determined by albedo of dry and wet soils, adjusted by an empirical coefficient $k_a$ (**Equation
(A22)**). Snow albedo was determined by snow age, as well as cumulative air temperature since the latest
snowfall, adjusted by the minimum snow albedo $a_{min}$ (**Equation (A23)**).

*3.5 Soil freezing point depression function development*

To solve the coupled water and heat flow equations, we needed a relation between soil temperature
and soil liquid water, i.e. soil freezing characteristics. In frozen soil, when soil temperature was below
zero, latent heat changed due to ice formation. When soil temperature continued decreasing, sensible heat
also changed. In CoupModel, we assumed that soil is totally frozen when temperature was below $T_f$ (-5
℃), when soil temperature was between 0 and $T_f$, sensible heat in soil was calculated as:
$$H = E(1 - \frac{L_f(w - \Delta z d_1 \theta_{wilt} \rho_w)}{E_f})(1 - r) \tag{16}$$



where $E$ is total heat stored in soil (J); $L_f$ is latent heat of freezing (J kg$^{-1}$); $w$ is water stored in soil (kg);
$\Delta z$ is soil thickness (m); $d_1$ is a factor accounting for the fraction of unfrozen water to soil wilting point
water content; $\theta_{wilt}$ is the wilting point water content when the pF value of soil water is 4.2 (m$^3$ m$^{-3}$); $\rho_w$
is density of water (kg m$^{-3}$); $E_f$ is energy when soil is totally frozen ($C_f T_f - L_f w_{ice}$, i.e. when soil
temperature is $T_f$, $C_f$ is heat capacity of frozen soil, J kg$^{-1}$ $^{\circ}$C$^{-1}$); $r$ is freezing point depression.
In modeling of soil frost, when soil was totally frozen at -5 $^{\circ}$C, the liquid water content was
determined by wilting point of soil ($\Delta z d_1 \theta_{wilt} \rho_w$), and adjusted by a coefficient $d_1$, as depicted in
**Equation (16)**. Ice content in soil was calculated as:

$$
\theta_i = \begin{cases} 0, T > T_0 \\ \frac{E-H}{\Delta z L_f \rho_{ice}}, T_f < T \le T_0 \\ \theta - d_1 \theta_{wilt}, T \le T_f \end{cases} \qquad (17)
$$

where $E$ is total energy stored in soil (J); $H$ is total energy stored in soil (=$C_f T$, J); $L_f$ is latent heat of
freezing (J kg$^{-1}$); $\Delta z$ is soil thickness (m); $d_1$ is a factor accounting for the fraction of unfrozen water to
soil wilting point water content; $\theta_{wilt}$ is the wilting point water content when the pF value of soil water is
4.2 (m$^3$ m$^{-3}$); $\rho_{ice}$ is density of ice (kg m$^{-3}$).
In CoupModel, the freezing-point depression was related to soil heat storage as below:

$$
r = \left(1 - \frac{E}{E_f}\right)^{d_2 \lambda + d_3} \min\left(1, \frac{E_f - E}{E_f + L_f w_{ice}}\right) \qquad (18)
$$

where $d_2$, $d_3$ are empirical constants; $\lambda$ is the pore size distribution index; $w_{ice}$ is water available for
freezing, kg, i.e. ($w - \Delta z d_1 \theta_{wilt} \rho_w$) in **Equation (18)**; $E_f$ is soil heat storage when soil temperature is $T_f$
($C_f T_f - L_f w_{ice}$), J.



In saline frozen soil, ice formation does not start at 0 $^{o}$C, but below 0 $^{o}$C. Freezing point $T_0$ (**Equation**
**(19)**) is a parameter related to soil type, salt type and salt content. In CoupModel, $T_0$ was assumed as 0 $^{o}$C,
which was not suitable for saline soils. In this study, two methods were implemented to consider salt
influences on freezing point depression. The first one was to set freezing point $T_0$ as a parameter in the
model, and this parameter could be determined by experiments on freezing point of different saline soils.
The second method was to relate $T_0$ to osmotic potential (**Equation (19)**). According to Banin and
Anderson (1974), the relationship between freezing point and salt solution could be written as below:
$$T_0 = -10^{-4+sc} \times \frac{\pi}{1.221} \tag{19}$$

where $T_0$ is the freezing point ($^{o}$C); $\pi$ is osmotic potential (in unit cm); $sc$ is a scale factor for
considering the influences of salt types on the relationship (range from -2 to 2); -4 is a constant for
converting osmotic potential unit from cm to MPa.
Soil salt and soil heat and water transport as well as soil freezing/thawing was connected by **Equation**
**(19)** with osmotic potential $\pi$, and freezing point would change as soil temperature and soil salt
concentration changed during simulation, since osmotic potential was determined by both soil
temperature and salt concentration:
$$\pi(z) = R(T + 273.15)\frac{c_{Cl}(z)}{M_{Cl}} \tag{20}$$

where $R$ is gas constant; $T$ is soil temperature (K); $c_{Cl}$ is salt concentration (kg m$^{-3}$); $M_{Cl}$ is mole mass of

Cl (35.5 g mol$^{-1}$).
*3.6 Calibration approach*
The sensitivity analysis and model calibration procedures are summarized in **Fig. 2**. We selected 58
parameters that have either shown a large influence on modeled water and heat dynamics in previous





studies and/or are known to be important in sensitivity analysis (cf. Gustafsson et al., 2001; Wu et al.,
2011; Metzger et al., 2015). The 58 parameters represented the major processes related to soil water, heat,
radiation as well as salt transport, 19 related to soil water process, 8 related to soil heat process, 19 related
to soil salt process, and 12 related to energy balance process (**Table A1**). We noted that 58 parameters
made the calibration very inefficient, since some of the parameters were assigned to different layers and
some were not so sensitive in comparison with others. We thus conducted a two-step calibration, with the
first step to find out the most important parameters from different model processes based on sensitivity
analysis, and the second step to calibrate the important parameters.
In the first step, the 58 parameters were tested for each site with 70000 simulations based on Monte
Carlo sampling method. Each of the simulations was run with randomly selected parameter values, thus
creating 70000 realizations. The most sensitive model parameters were then identified for each site based
on their relative importance on performance metrics (e.g. $R^2$, determination coefficient between
simulation and observations, and ME, the mean deviations between simulation and observations). This
was done by using the LGM (Lindeman, Gold and Merenda) method (Lindeman et al., 1980) that
averages the sequential sums of squares over all orderings of regressors, which calculates the relative
importance of each parameter on model performance metrics and ranks them. Based on the ranking of
parameters, 8 to 11 sensitive parameters (i.e. 3 common parameters for two sites, another 5 for site
Qianguo and another 8 for site Yonglian) were then selected in the second step with 10000 simulations
for each site. It is important to note that the sensitive parameters may be different from site to site
depending on site-specific characteristics, although initial parameters and their ranges were equivalent for
all sites.





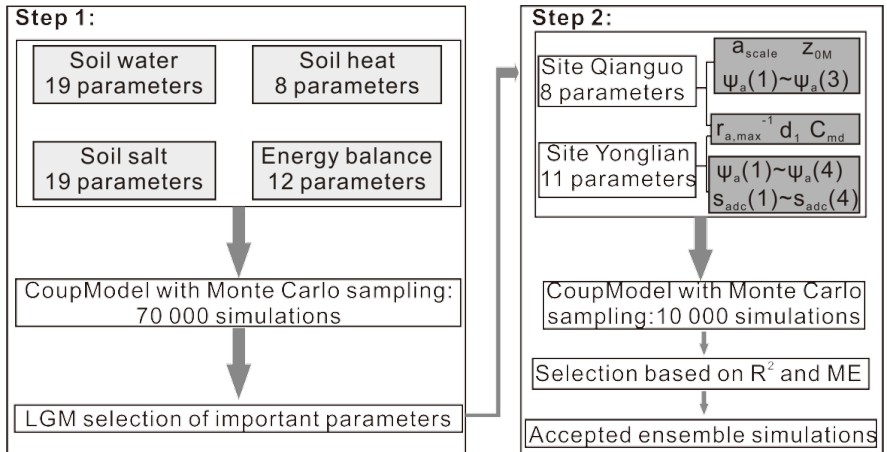

**Fig. 2** Model sensitivity analysis and calibration procedures. Step 1 denotes procedures for selecting
important parameters with LGM method, step 2 represents model calibration and selection of acceptable
simulations.

From the 10000 simulations of the second calibration step, ensemble of parameter sets for each site

was then selected based on statistical performance metrics (determination coefficient $R^2$ and mean error
ME) for temperature and water at several depths (**Table 2**). In addition to useful information about site-
specific processes and their representations in the model, the ensemble simulation results (9 for site
Qianguo, 16 for site Yonglian) simulated using accepted parameter sets were used for analysis water,
energy and salt balance over the simulation period.
**Table 2** Criteria applied to model performance metrics in selection of behavioral simulations.

| Site name | Qianguo | | | | Yonglian | | | |
|---|---|---|---|---|---|---|---|---|
| | $R^2$ | | ME | | $R^2$ | | ME | |
| Soil depth | T | $\theta$ | T ($^{o}$C) | $\theta$ (%) | T | $\theta$ | T ($^{o}$C) | $\theta$ (%) |
| 5cm | [0.85,1] | [0.3,1] | [-1,1] | [-5,5] | [0.9,1] | [0.5,1] | [-0.5,0.5] | [-3,3] |
| 15 cm | [0.9,1] | [0.8,1] | [-0.5,0.5] | [-5,5] | [0.9,1] | [0.5,1] | [-0.5,0.5] | [-3,3] |
| 25 cm | [0.9,1] | [0.5,1] | [-0.5,0.5] | [-3,3] | [0.9,1] | [0.5,1] | [-0.5,0.5] | [-3,3] |
| 35 cm | [0.85,1] | [0.7,1] | [-0.5,0.5] | [-2,2] | [0.9,1] | [0.5,1] | [-0.5,0.5] | [-3,3] |
| [a]GWTD | / | / | / | / | [0.6,1] | | | [b][-0.1,0.1] |

[a]GWTD is ground water table depth
[b]unit for GWTD is m




## 4. Results and Discussion

### 4.1 Freezing point depression

In **Fig. 3** and **Fig. 4**, the sensitivity of model to freezing point depression is analyzed at 5 cm depth (the same was done for other soil depths, not shown here), based on model results from one of the behavioral simulations at each site. The influences of freezing point on soil heat are obvious (**Fig. 3**). When soil freezing temperature changed from 0 $^{\circ}$C to below zero, the relationship between soil temperature and soil heat storage changed accordingly. The model performance was improved when freezing point depression was related to soil salt. Mean error (ME) for soil temperature at 5 cm depth decreased from 1.25 $^{\circ}$C to 0.29 $^{\circ}$C (improved by 77%) at site Qianguo, and from 2.54 $^{\circ}$C to 1.83 $^{\circ}$C (improved by 28%) at site Yonglian, when freezing point decreased from 0 $^{\circ}$C to -3 $^{\circ}$C.

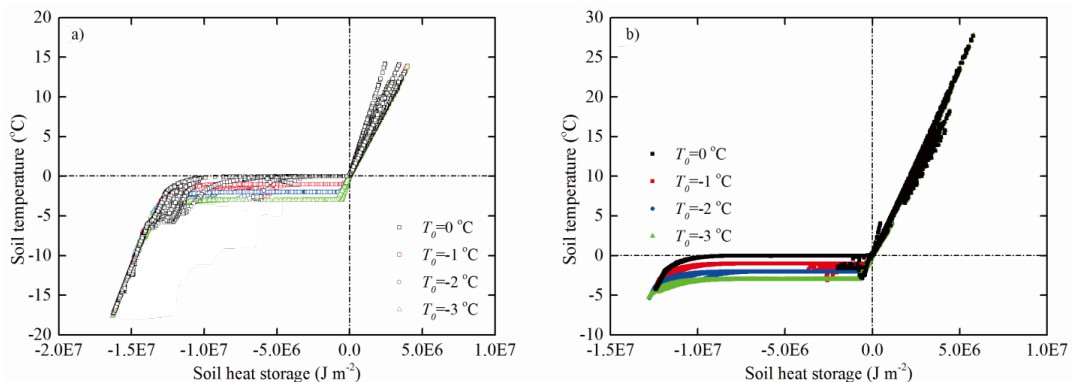

**Fig. 3** Relationships between soil temperature and soil heat storage derived from modeling for different freezing points at 5 cm depth at a) site Qianguo and b) site Yonglian.

The relationships between soil temperature and soil heat storage at 5 cm depth were different when various *sc* values were assigned (**Fig. 4**). This indicated that different types of salt also influence soil freezing/thawing. Meanwhile, ME decreased from 1.35 $^{\circ}$C to 0.64 $^{\circ}$C (improved by 53%) when *sc* changed from 0 to 1 at site Qianguo. At site Yonglian, ME decreased from 2.54 $^{\circ}$C when *sc* is 0 to 2.14





$^{o}$C (improved by 16%) when $sc$ was 1. This indicated that in determination of freezing point, not only the
salt content, but also the salt type should be taken into consideration for reducing uncertainty in modeling
soil temperature.

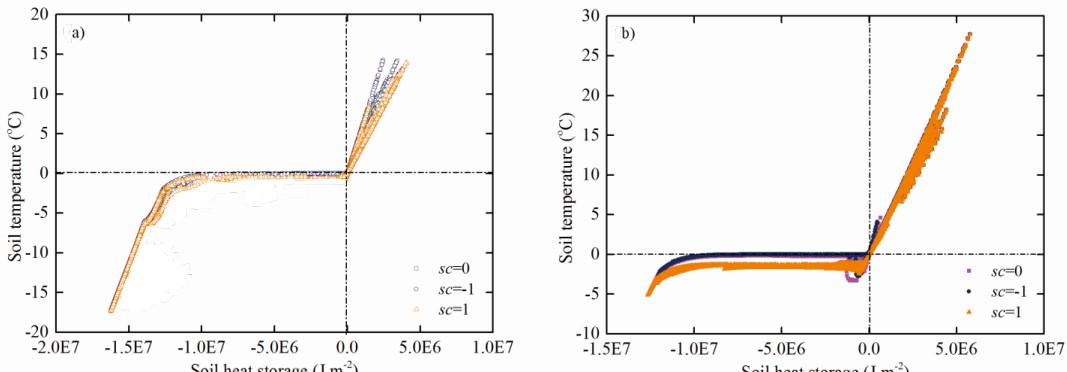

**Fig. 4** Relationships between soil temperature and soil heat storage derived from modeling for different $sc$
values at 5 cm depth at a) site Qianguo and b) site Yonglian.
*4.2 LGM relative importance in calibration parameters*
At site Qianguo, for soil temperature $R^2$ at 4 depths (**Fig. 5** a)-d)), $z_{oM,snow}$ (momentum roughness
length of snow), which is to estimate surface aerodynamic resistance, was found to be most important.
This parameter would influence surface energy balance and eventually impact heat transport in soil
profile. The other important parameter for soil temperature $R^2$ at 15, 25 and 35 cm depths was $C_{md}$, which
is a parameter to adjust thermal conductivity of surface frozen layer.
For liquid water content $R^2$ at 4 depths (**Fig. 5** e)-h)), the most important parameters were $z_{0M,snow}$,
$r_{a,max}^{-1}$, $d_1$ and $\psi_a$ of different depths that were related to energy balance and soil heat and water transport.
$z_{0M,snow}$ was already detected to be important for soil temperature. This indicated that surface energy
balance in snowpack at site Qianguo is important for both soil heat and water transport.

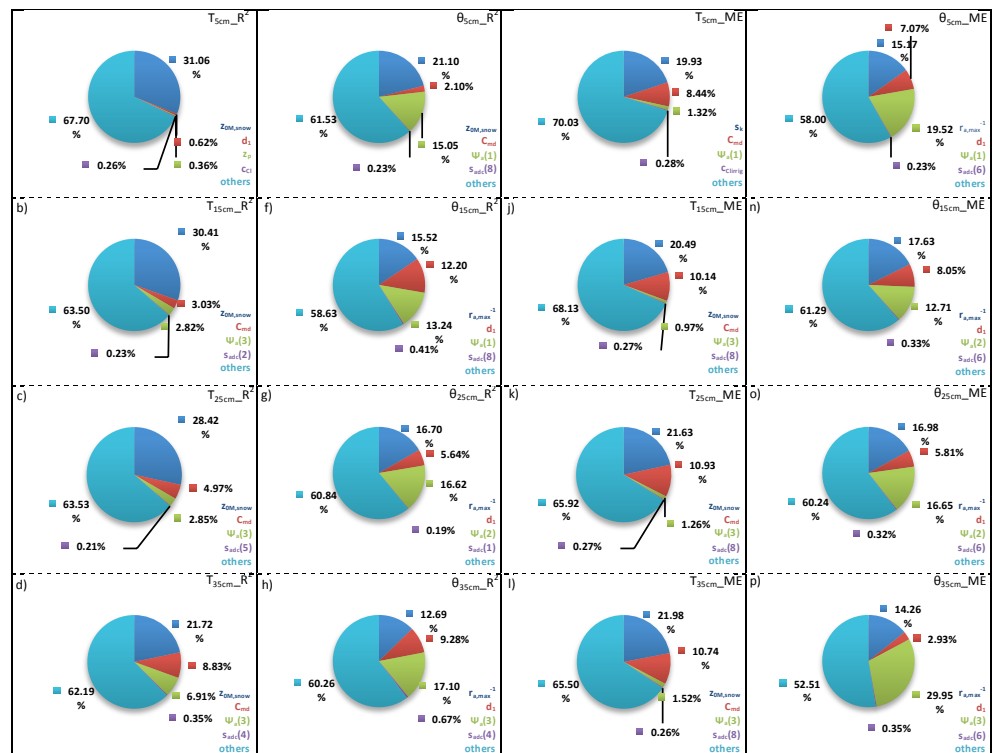

**Fig. 5** Relative importance for parameters to $R^2$ a)-h) and ME i)-p) of soil temperature and soil water at 4 depths (5, 15, 25, 35 cm) at site Qianguo. Parameters shown in each sub-figure are the most important parameter from each group, i.e. energy balance, soil heat, soil water, soil salt, and the other parameters.

For soil temperature ME at site Qianguo (**Fig. 5** i)-l)), important parameters were similar to soil temperature $R^2$, except at 5 cm depth, with $s_k$ showing the greatest importance (**Fig. 5** i)). $s_k$ is for estimate of snow thermal conductivity, and determines energy balance in snowpack. Site Qianguo was covered by snow during soil freezing, thus accurate estimates of snow energy balance would help in improving model performance on soil temperature. At site Qianguo, important parameters for soil liquid water content ME were similar to $R^2$, showing that parameters related to surface energy balance ($r_{a,max}^{-1}$), soil freezing characteristics ($d_1$) and soil water characteristics ($\psi_a$) have great importance to soil water transport.



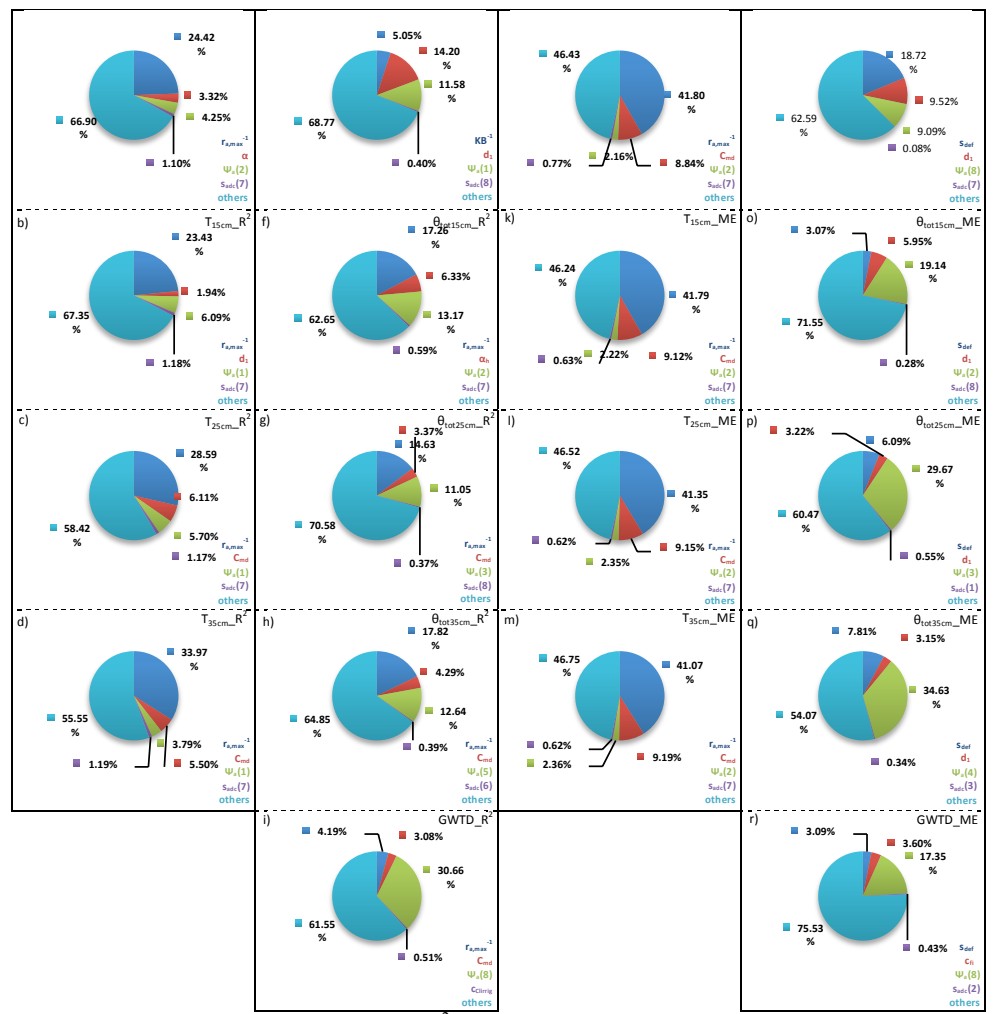

**Fig. 6** Relative importance for parameters to $R^2$ a)-i) and ME j)-r) of soil temperature and soil water at 4

depths (5, 15, 25, 35 cm) as well as groundwater at site Yonglian. Parameters shown in each sub-figure

are the most important parameter from each group, i.e. energy balance, soil heat, soil water, soil salt, then

the other parameters.

At site Yonglian, $r_{a,max}^{-1}$ is shown to be the most important parameter to soil temperature $R^2$ at 4 depths

(**Fig. 6** a)-d)). The other important parameters were related to soil frost and soil water characteristics (e.g.

$\alpha, d_1, C_{md}, \psi_a$). For soil total water content and groundwater table depth $R^2$, the important parameters

from different groups were also important to soil temperature $R^2$. We can see that except for $\psi_a$ at various



depths, the most important parameters were related to energy balance and soil heat transport in frozen
soils (e.g. $kB^{-1}$, $r_{a,max}^{-1}$, $d_1$, $C_{md}$, $\alpha_h$) (**Fig 6** e)-i)). For soil temperature ME (**Fig. 6** j)-m)), the same
parameters were shown important as soil temperature $R^2$. For soil total water content and groundwater
table depth ME (**Fig. 6** n)-r)), the most important parameters were $s_{def}$, $d_1$ and $\psi_a$ from different soil
depths. $s_{def}$ is the maximum soil surface water deficit for calculation of surface water balance and
adjusting soil surface vapour pressure. It determines the estimate of soil evaporation. The great
importance of $s_{def}$ to soil water ME indicated that at site Yonglian, soil evaporation is important in soil
water transport.
Soil salt related parameters did not show that great importance (around 1% relative importance) to soil
temperature and soil water at two sites. Even we have developed a new relationship between osmotic
potential and soil freezing temperature and it has shown to be able to improve model performance on
simulation of soil temperature, the parameter $sc$ did not show great importance at two sites. This
indicated that for two sites, $sc$ could be assigned as fixed values for different types of salt. We only
noticed the adsorption coefficients of salt at various layers show some importance to soil temperature and
water. This was because they determine the osmotic potential of soil water and thus impact soil heat and
water transport simulation.

*4.3 Prior and posterior parameters*

In **Table 3**, the important parameters and their posterior ranges at two sites are depicted. The posterior
mean value of $r_{a,max}^{-1}$ was reduced to 1/3 of prior mean value, and mean value for $d_1$ was also reduced
from prior at site Qianguo. The R$_{ratio}$ (range ratio, defined as the posterior range width ratio to prior range
width) for $r_{a,max}^{-1}$ and $d_1$ was 0.12 and 0.18, respectively for site Yonglian (**Table 3**). Parameters
$z_{0M,snow}$, $a_{scale}$ and $C_{md}$ at site Yonglian had larger posterior mean values than prior, and the R$_{ratio}$ was
0.54 and 0.68, respectively. Parameters $\psi_a(1)$ to $\psi_a(3)$ at site Yonglian also obtained generally larger
posterior mean values after calibration, with R$_{ratio}$ of 0.18 to 0.50.



**Table 3** Important parameters and their accepted ranges at site Qianguo and Yonglian.

| Parameters | Explanation | Posterior values | | Accepted runs mean | [a]R$_{ratio}$ |
| --- | --- | --- | --- | --- | --- |
| | | Minimum | Maximum | | |
| $r_{a,max}^{-1}$ | Minimum turbulent exchange coefficient for bare soil in **Equation (A18)** (Jordan, 1991) | [b]0.51(0.01) | 0.57(0.04) | 0.54(0.02) | 0.12(0.58) |
| $d_1$ | Fraction of unfrozen water to wilting point when soil temperature is at -5 °C in **Equation (16)** | 0.40(0.23) | 0.48(0.45) | 0.43(0.34) | 0.18(0.39) |
| $C_{md}$ | Maximum frozen soil thermal conductivity damping coefficient in **Equation (A12)** | 0.67(0.80) | 0.87(0.89) | 0.76(0.86) | 0.52(0.22) |
| $z_{0M,snow}$ | Momentum roughness length of snow (m) in **Equation (A16)** | 0.02 | 0.04 | 0.03 | 0.54 |
| $a_{scale}$ | Matric water adsorption coefficient for calculation of $s_{mat}$ in **Equation (A2)** | 2.78 | 9.55 | 6.36 | 0.68 |
| $s_{adc}(1)$ | Adsorption coefficient of salt in **Equation (11)** | (0.04) | (0.50) | (0.24) | (0.93) |
| $s_{adc}(2)$ | Adsorption coefficient of salt in **Equation (11)** | (0.02) | (0.49) | (0.21) | (0.91) |
| $s_{adc}(3)$ | Adsorption coefficient of salt in **Equation (4)** | (0.10) | (0.50) | (0.33) | (0.74) |
| $s_{adc}(4)$ | Adsorption coefficient of salt in **Equation (4)** | (0.01) | (0.48) | (0.24) | (0.76) |
| $\psi_a(1)$ | Air entry value of soil (%) in Brooks & Corey equation in **Equation (A3)** | 38.34(30.94) | 89.91(97.24) | 57.71(65.12) | 0.46(0.81) |
| $\psi_a(2)$ | Air entry value of soil (%) in Brooks & Corey equation in **Equation (A3)** | 26.49(15.94) | 97.00(93.72) | 80.36(48.49) | 0.18(0.78) |
| $\psi_a(3)$ | Air entry value of soil (%) in Brooks & Corey equation in **Equation (A3)** | 40.83(30.60) | 92.27(99.02) | 68.83(64.46) | 0.50(0.66) |
| $\psi_a(4)$ | Air entry value of soil (%) in Brooks & Corey equation in **Equation (A3)** | (27.33) | (63.93) | (41.18) | (0.36) |

[a]Rratio ratio of posterior parameter range to prior parameter range
[b]value without brackets is for site Qianguo, value with brackets is for site Yonglian

For $C_{md}$ at site Yonglian, the R$_{ratio}$ was 0.22, much smaller than at site Qianguo (0.52). R$_{ratio}$ of $d_1$ at

site Yonglian was 0.39, larger than at site Qianguo (0.18). Parameter $r_{a,max}^{-1}$ at site Yonglian showed
totally different R$_{ratio}$ and posterior mean values from site Qianguo, with R$_{ratio}$ of 0.58 and posterior mean
value of 0.02, respectively. As discussed above, $C_{md}$, $d_1$ and $r_{a,max}^{-1}$ were important parameters at both
sites. They controlled soil heat and energy balance and can influence soil freezing/thawing. Salt
adsorption coefficient $s_{adc}$ at four depths did not show large changes between posterior and prior



distributions, with $R_{ratio}$ from 0.74 to 0.93. The $R_{ratio}$ of $\psi_a$ from four depths at site Yonglian varied from
0.36 to 0.81. The large differences in posterior ranges of these parameters indicated these two sites have
different surface water and energy balance situations. Site Qianguo is more humid in winter and has more
snow events, while site Yonglian has very dry winter but more salt influences on freezing/thawing due to
higher salinity at this site. At site Qianguo, parameters such as $z_{0M,snow}$ and $a_{scale}$ also showed to be
important, and $s_{adc}$ at various depths at site Yonglian were shown to be very sensitive.
*4.4 Soil temperature and water*
At site Qianguo, soil temperature and soil liquid water content were measured manually at daily
resolution due to difficulties in installing automatic measurement instruments at farmers' land. Accepted
simulations generally can capture soil temperature and water dynamics and can cover the observations
within their ranges (**Fig. 7** a)-b)). After calibration, the mean value of $R^2$ for soil temperature and soil
liquid water content at 5 cm depth was 0.87 and 0.31, respectively. Meanwhile, the mean values of ME
for soil temperature and soil liquid water content at 5 cm depth was -0.41 °C and -4.89%, respectively.
At site Yonglian, hourly soil temperature was obtained in calibration, and achieved high $R^2$ for soil
temperature, with mean $R^2$ of 0.90 at 5 cm depth. However, soil temperature was underestimated from
end of November to middle of January (**Fig. 7** c)). This was mainly due to ice coverage at site Yonglian
during this period. After flooding irrigation at the beginning of November, water ponding in the field (~10
cm water) was rapidly frozen and kept covering soil surface until middle of January. For this period, ice
coverage disturbed water and energy balance at the site Yonglian. Even the snowpack was considered at
site Yonglian and a detailed scheme for snow water and energy balance was illustrated in CoupModel, it
obviously cannot describe ice coverage in our case. In the future development of the CoupModel, we
recommended inclusion of a new scheme for water and energy balance on ice coverage.



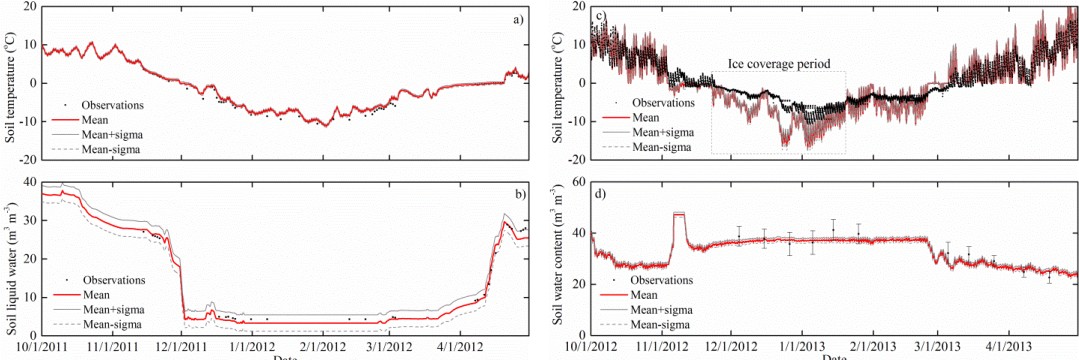

**Fig. 7** Comparison of soil temperature and soil water at 5 cm depth at two sites. a)-b) at site Qianguo, c)-d) at site Yonglian. Red line for mean of behavioral simulations, grey lines for simulated mean values +/- standard deviation (sigma), error bar in d) for standard deviation of observations from various plots.

Due to failure of TDR in measuring water in salinized frozen soil, we only sampled total water content at site Yonglian. Soil total water content at 5 cm depth had good performance with mean $R^2$ from accepted simulations of 0.80. Even with only 14 sampling dates from each of five plots were selected, the calibrated model can capture soil water dynamics well. Nevertheless, we also noticed large variations in measured total water content from different plots, as indicated by error bars in **Fig. 7** d). In the future development of soil water measurement methods, it is necessary to introduce more accurate measurement methods for soil water in saline frozen soils in order to obtain consistent observations of soil water dynamics during winter.

*4.5 Model validation on water and salt storage*

Comparison of simulated water storage with measured water storage at different soil depths is depicted in **Fig. 8**. Results indicated that CoupModel could predict water process well in upper 40 cm soil layer, but some large deviations mainly occurred for 40 to 100 cm at two sites between simulated and observed soil water storages. This was because the accepted simulations was derived by constraining model performance for variables (soil temperature and soil water) in upper 40 cm soil layer, and the data from 40-100 cm depth was not used for calibration. This indicated that there might be some unforeseen





processes in the lower soil layer, and they would influence water processes in whole soil profile (from
surface to groundwater). Since the calibration was focusing on the surface water and energy balance, and
the upper layer water process was shown to be well-represented by the model, the more detailed
consideration of lower layer water processes exceeded the scope of this study. Further work would
include calibration of the model in the whole soil profile with more detailed measurements.
**Fig. 9** shows the simulated salt storage in comparison with measured salt storage based on measured
data at various soil depths. At site Qianguo, the Br⁻ storage was generally overestimated by the model in
comparison with measured Br⁻ storage in whole soil profile. The simulated Br⁻ storage showed larger
uncertainty than the measured. Similarly, at site Yonglian, the simulated Cl⁻ storage at various layers was
larger than measured Cl⁻ storage.

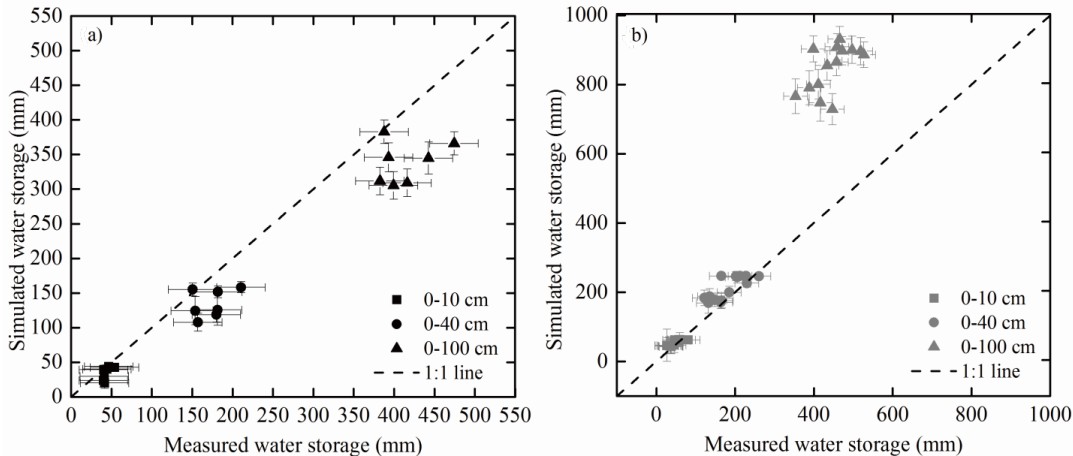

**Fig. 8** Comparison of simulated accumulated water storage with measured accumulated water storage at
two sites at various soil depths. a) site Qianguo and b) site Yonglian. X error bar denotes standard
deviation of observations from various plots, Y error bar denotes standard deviation of water storage from
behavioral simulations.





The overestimation of Cl⁻/Br⁻ storage at various depths indicated that the upward movement of salt
with water was over-estimated. This might be due to the neglecting of diffusion and expulsion of salt in
model. Cary and Mayland (1972) have shown that, the diffusion and expulsion processes in frozen soil
actually played important roles in salt transport even though the convection was the major process. This
was because when soil was frozen, soil solution concentrated. The concentration of soil solution would
increase salt concentration gradient between soil layers. In addition, high salt concentration at low
temperature would cause salt expulsion from solution due to low salt saturation (Wang et al., 2016).
However, it was difficult to measure the diffusion and expulsion of salt in frozen soil. More detailed
experiments on diffusion and expulsion of salt are necessary in study of water, heat and salt coupled
transport in frozen soils. Validation of soil water and salt storage more data on salt transport as well as
water transport would be of importance in calibration of model, since the water and salt transport
processes are tightly coupled.

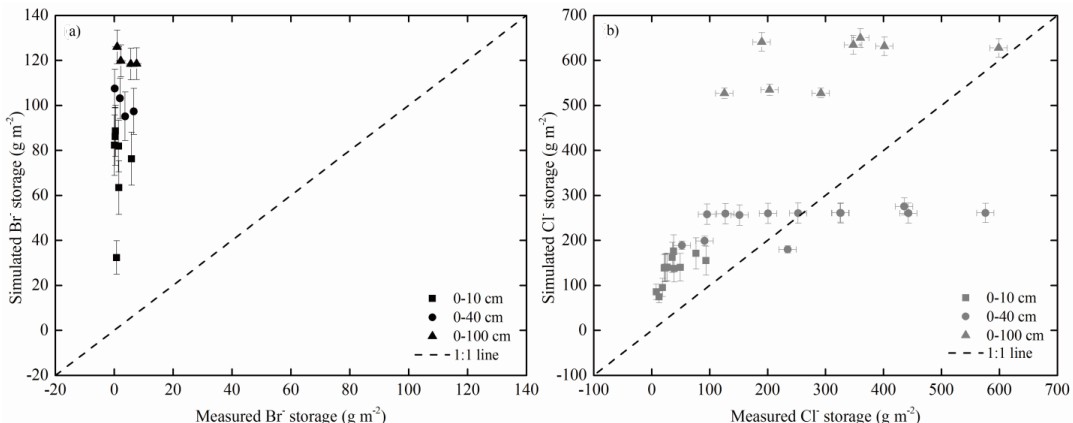

**Fig. 9** Comparison of simulated salt (Br⁻ and Cl⁻) storage with measured at two sites. a) site Qianguo and
b) site Yonglian. X error bar denotes standard deviation of observations from various plots, Y error bar
denotes standard deviation of salt storage from behavioral simulations.
**5.  Conclusions**





Water, heat and salt migrations are coupled in agricultural field. It is important to understand the
mechanisms behind this coupling for better water management in cold arid regions. In this study, water,
temperature and salt transport during freezing/thawing was simulated and compared with measurements
in two seasonally frozen soils in northern China. Uncertainties in both measurements and model were
evaluated using Monte-Carlo sampling method and a newly developed CoupModel including salt
influences on soil freezing point. Multiple criteria were applied to different model performance metrics
for selection of behavioral simulations in evaluating soil temperature, liquid or total water content and
groundwater table. With the new freezing point determination methods, simulated soil temperature
performance was improved with respect to mean error (ME), by 16% to 77%. Parameters determining
energy balance at soil surface as well as soil freezing characteristics were shown important for modeling
soil water and heat transport processes in LGM analysis. Parameters such as $r_{a,max}^{-1}$, $d_1$ and $C_{md}$ had large
differences in posterior distributions from prior distributions, with posterior range ratio (R$_{ratio}$) of 0.12 to
0.52 and 0.22 to 0.58 at site Qianguo and site Yonglian, respectively. Calibration of the important
parameters has improved model performance a lot, with mean posterior $R^2$ values for soil temperature of
0.87 and 0.90, and mean posterior $R^2$ values for soil water (liquid and total) of 0.31 and 0.80, at site
Qianguo and Yonglian, respectively. Validating of the calibrated model results against soil water and salt
storages at different depths has shown that soil water storage was well represented at upper soil layers
form surface to 40 cm depth, with water storage at 40-100 cm depth at site Qianguo underestimated, and
water storage at 40-100 cm depth overestimated. Meanwhile, salt storage at two sites were generally
overestimated by the model in the whole 0-100 cm soil profile, mainly due to lack in considering more
salt transport processes such as diffusion and expulsion in frozen soils. The study has emphasized that
taking influences of salt on freezing point depression into account in CoupModel can improve model
performance and reduce modeling uncertainty. But detailed experiments and model development on salt
transport mechanism (e.g. diffusion and expulsion of salt in frozen soils) would be very necessary in
investigation of salinization and in water management in cold arid agricultural regions.





## Acknowledgements

This research was funded by the National Key Research and Development Program of China (Grant Nos. 2017YFC0403304, 2016YFC0501304, 2016YFC0400203), Major Program of National Natural Science Foundation of China (Grant Nos. 51790532, 51790533) and Open Foundation of State Key Laboratory of Water Resources and Hydropower Engineering Science (No. 2017NSG02). We would like to thank Ms. Ai'ping Chen and Mr. Yang Xu from the Yichang Experimental Site for supplying the meteorological data of the studied sites. The authors also appreciate Mr. Pengju Yang, Mr. Dacheng Li and Mr. Weixing Quan for helping analyze the soil samples and processing data in laboratory.

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

**Appendix**
**Table A1** Calibrated parameters with their prior ranges.

| Model parameters | Descriptions | Symbol | [c]Equation | Prior Min | Prior Max |
|---|---|---|---|---|---|
| *Soil water processes calibrated parameters* | | | | | |
| DvapTortuosity | Tortuosity of vapour | $d_{vapb}$ | A1 | 0.01 | 2 |
| AScaleSorption | Soil matric water adsorption coefficient | [a]$a_{scale}$ | A2 | 0.1 /[b]0.02 | 10 /[b]0.1 |
| Air Entry(1)-Air Entry(8) | Air entry value of soil (%) | $\psi_a$ | A3 | 0.01 | 100 |
| HighFlowDampC | Damping coefficient for hydraulic conductivity in high-flow domain (%) | $c_{\theta,i}$ | A5 | 0.1 /[b]0.1 | 80 /[b]50 |
| LowFlowCondImped | Impedance coefficient for hydraulic conductivity in low-flow domain due to ice existence | $c_{fi}$ | A6 | 0.1 | 10 |
| MinimumCond | Minimum hydraulic conductivity for | [a]$k_{min\,uc}$ | A7 | 0.0001 | 0.001 |



| | | | | | |
|---|---|---|---|---|---|
| Value | determining hydraulic conductivity change with soil temperature (mm d$^{-1}$) | | | [b]1.0E-06 | [b]1.0E-05 |
| SurfPoolMax | Maximum soil surface pool (mm) | $w_{pmax}$ | A8 | 20 | 100 |
| DrainSpacing | Distance between two drainage tubes (m) | $d_p$ | A9 | 250 | 300 |
| DrainLevel | Initial drainage level (m) | $z_p$ | A9 | -2.5 | -2 |
| EmpGFLevPeak | Empirical groundwater level peak value (m) | $z_1$ | A10 | -1.5 | -0.5 |
| EmpGFlowPeak | Empirical groundwater flow peak value (m) | $q_1$ | A11 | 5 | 15 |
| InitialGroundWater | Initial groundwater table depth (m) | - | - | -1 [b]-2.2 | -0.5 [b]-1.8 |
| DrainLevelMin | Minimum drainage level (m) | - | - | -1.5 | -0.5 |
| *Soil heat processes calibrated parameters* | | | | | |
| Ballard_Arp Alpha | Coefficient for calculation of thermal conductivity with Ballard & Arp method | $\alpha$ | A11 | 0.1 | 0.3 |
| Ballard_Arp Beta | Coefficient for calculation of thermal conductivity with Ballard & Arp method | $\beta$ | A11 | 10 | 30 |
| Ballard_Arp a | Coefficient for calculation of thermal conductivity with Ballard & Arp method | $a$ | A11 | 0.4 | 0.6 |
| CfrozenMaxDamp | Maximum damping coefficient of frozen soil thermal conductivity | $C_{md}$ | A12 | 0.5 | 0.9 |
| AlphaHeatCoef | Heat transfer coefficient for re-freezing of water in high-flow domain (W m$^{-1}$ $^o$C$^{-1}$) | [a]$\alpha_h$ | A13 | 0.1 [b]500 | 5000 [b]5000 |
| FreezepointFWi | Coefficient for determining liquid water content when soil is frozen at -5 $^o$C | $d_1$ | Equation(16) | 0.1 | 0.5 |
| *Soil salt processes calibrated parameters* | | | | | |
| SaltInitConc | Initial salt concentration (mg L$^{-1}$) | [a]$c_{Cl}$ | - | 20 [b]800 | 40 [b]1200 |
| SaltInputConc | Salt concentration from precipitation input (mg L$^{-1}$) | [a]$c_{Cldep}$ | - | 1 [b]0.01 | 10 [b]500 |
| SaltIrrigationConc | Salt concentration from irrigation (mg L$^{-1}$) | [a]$c_{Clirrig}$ | - | 1500 [b]500 | 2500 [b]1000 |
| Ad_c(1)-Ad_c(16) | Salt adsorption coefficient | $s_{adc}$ | - | 0 | 0.5 |
| *Energy balance processes calibrated parameters* | | | | | |
| EquilAdjustPsi | Coefficient for adjustment of vapor difference between upper soil and surface | $\psi_{eg}$ | A14 | 0.5 | 1.2 |
| MaxSurfExcess | Maximum soil surface water excess in surface water balance estimate (mm) | $s_{excess}$ | A15 | 0.5 | 2 |
| MaxSurfDeficit | Soil surface maximum water deficit in surface water balance estimate (mm) | $s_{def}$ | A15 | -3 | -1 |
| RoughLBareSoilMom | Momentum roughness length of bare soil (m) | [a]$Z_{0M}$ | A16 | 1.0E-05 | 0.05 |
| RoughLMomSnow | Momentum roughness length of snow | [a]$z_{0M,snow}$ | A16 | 0.005 [b]0.025 | 0.05 [b]0.05 |
| KBMinusOne | Ratio of momentum and heat roughness length | $kB^{-1}$ | A17 | 0 | 2.5 |
| WindLessExchangeSoil | Windless exchange coefficient of bare soil | $r_{a,max}^{-1}$ | A18 | 0.5 [b]1.0E-04 | 1 [b]0.05 |
| MaxSoilCondens | Maximum soil water condensation rate (mm d$^{-1}$) | $e_{max,cond}$ | A19 | 2/[b]1 | 4/[b]2 |
| KonzelmannCoef_1 | Konzelmann coefficient for estimate of longwave radiation | $r_{kl}$ | A20 | 0.15 | 0.31 |
| SThermalCond | Snow thermal conductivity coefficient (W | [a]$s_k$ | A21 | 2.5E-06 | 1.0E-05 |





| Coef | m$^5$ $^o$C$^{-1}$ kg$^{-2}$) | | | [b]1.0E-06 | [b]1.0E-05 |
|---|---|---|---|---|---|
| AlbedoKExp | Exponential coefficient for estimate of soil albedo | $k_a$ | A22 | 0.5 | 1.5 |
| AlbSnowMin | Minimum snow albedo | $a_{min}$ | A23 | 30 | 50 |

[a]This parameter was sampled using stochastic log distribution, the others using stochastic linear distribution
[b]Range specific for site Yonglian
[c]Equation is corresponded to the number in **Table A2** or in main text

**Table A2** Calibrated parameters related equations and their descriptions in CoupModel.

| Equation no. | Equations | Descriptions |
|---|---|---|
| | *Soil water processes* | |
| A1 | $D_v = d_{vapb} f_a D_0$ <br><br> $D_0 = \left(\dfrac{T + 273.15}{273.15}\right)^{1.75}$ <br><br> $c_v = \dfrac{M_{water} e_v}{R(T + 273.15)}$ <br><br> $e_v = e_s e^{\left(\frac{-\psi M_{water} e_v}{R(T+273.15)}\right)}$ <br><br> where $f_a$ is the fraction of air-filled pores, $D_0$ is the diffusion coefficient in free air (m$^2$ s$^{-1}$) and $d_{vapb}$ is a parameter accounting for tortuosity and the enhancement of vapor transfer observed in measurements compared with theory, $\psi$ is matric potential (m, with positive value), $M_{water}$ is mole mass of water (18 g mol$^{-1}$), $R$ is gas constant, $e_s$ is saturated vapor pressure (m), $T$ is soil temperature (K). | Soil vapor diffusion coefficient |
| A2 | $S_{mat} = a_{scale} a_r k_{mat} pF$ <br><br> where $k_{mat}$ is matric hydraulic conductivity (m s$^{-1}$), $a_r$ is the ratio between compartment thickness, $\Delta z$, and the unit horizontal area represented by the model, pF is $\log_{10} \psi$, $a_{scale}$ is an empirical scaling coefficient accounting for the geometry of aggregates. | Sorption capacity rate estimation |
| A3 | $S_e = \left(\dfrac{\psi}{\psi_a}\right)^{-\lambda}$ <br><br> where $\psi_a$ is the air-entry tension (m), $\lambda$ is the pore size distribution index and $S_e$ the effective saturation. | Water retention curve defined by Brooks and Corey (1964) |
| A4 | $k_w^* = k_{mat} S_e^{(n+2+2/\lambda)}$ <br><br> where $k_{mat}$ is matric hydraulic conductivity (m s$^{-1}$), $\lambda$ is the pore size distribution index, $S_e$ the effective saturation and $n$ is tortuosity. | Unsaturated soil hydraulic conductivity calculated by Mualem (1976) equation |
| A5 | $k_{fh} = e^{-\frac{\theta_i}{c_{\theta,i}}} \left(k_w(\theta_{tot}) - k_w(\theta_{lf} + \theta_i)\right)$ <br><br> where $k_w(\theta_{tot})$ is hydraulic conductivity for pores saturated with water (m s$^{-1}$), $k_w(\theta_{lf} + \theta_i)$ is hydraulic conductivity when water flow in low domain | Hydraulic conductivity in high flow domain |




| | | |
|---|---|---|
| | with ice existence (m s$^{-1}$), $\theta_i / c_{\theta,i}$ is reduced factor, and $c_{\theta,i}$ is impedance factor. | |
| A6 | $$k_{wf} = 10^{-c_{fi}Q} k_w$$ where $c_{fi}$ is impedance factor, and $Q$ is heat quality, as a ratio of ice content to total water content. | Hydraulic modification for low flow domain |
| A7 | $$k_w = (r_{AOT} + r_{A1T}T_s)\max(k_w^*, k_{minuc})$$ where $r_{AOT}$, $r_{A1T}$ ($^{o}$C$^{-1}$) and $k_{minuc}$ (m s$^{-1}$) are parameters. $k_w^*$ is the total hydraulic conductivity (m s$^{-1}$), as a sum of hydraulic conductivity from matrix and macro pores. | Actual unsaturated hydraulic conductivity after temperature correction |
| A8 | $$q_{surf} = a_{surf}\left(W_{pool} - w_{pmax}\right)$$ where $a_{surf}$ is an empirical coefficient, $W_{pool}$ is the total amount of water in the surface pool (m), and $w_{pmax}$ is the maximal amount of water stored on soil surface without causing surface runoff (m). | Surface runoff estimation |
| A9 | $$q_{wp} = \frac{4k_{s1}(z_{sat} - z_p)}{d_p^2} + \frac{8k_{s2}z_D(z_{sat} - z_p)}{d_p^2}$$ where $k_{s1}$, $k_{s2}$ are saturated hydraulic conductivities above and below water level (m s$^{-1}$), respectively; $z_D$ is soil depth below drainage level (m), $z_p$ is drainage depth to soil surface (m), $d_p$ is distance between two drainage tubes. | Hooghoudt drainage equation |
| A10 | $$q_{gr} = q_1 \frac{\max\left(0, z_1 - z_{sat}\right)}{z_1} + q_2 \frac{\max\left(0, z_2 - z_{sat}\right)}{z_2}$$ where $q_1$, $q_2$ are maximum and minimum empirical groundwater flow (m s$^{-1}$), respectively; $z_1$ and $z_2$ are highest and lowest empirical groundwater level (m), respectively. | Empirical drainage equation |
| *Soil heat processes* | | |
| A11 | $$K_{soil} = (K_{sat} - K_{dry})Ke + K_{dry}$$ $$K_{dry} = \frac{(aK_{solid} - K_{air})\rho_b + K_{air}\rho_p}{\rho_p - (1-a)\rho_b}$$ For unfrozen soils, $$Ke = \theta_{sat}^{0.5(1+V_{om,s}-\alpha V_{sand,s}-V_{cf,s})\left[\left(\frac{1}{1+\exp(-\beta\theta_{sat})}\right)^3 - \left(\frac{1-\theta_{sat}}{2}\right)^3\right]^{1-V_{om,s}}}$$ For frozen or partially frozen soils, $$Ke = \theta_{sat}^{1+V_{om,s}}$$ where $Ke$ is Kersten number (-), $\theta_{sat}$ is saturation (m$^3$ m$^{-3}$); $V_{om,s}$ is volumetric fraction of organic matter (-), $V_{sand,s}$ is volumetric fraction of sand (-), $V_{cf,s}$ is volumetric fraction of coarse fragments (-), $\alpha$ and $\beta$ are adjustment factor (-), $K_{solid}$ is solid thermal conductivity (W m$^{-1}$ $^{o}$C$^{-1}$), $K_{air}$ is air thermal conductivity (W m$^{-1}$ $^{o}$C$^{-1}$), $\rho_b$ is bulk density (kg m$^{-3}$), $\rho_p$ is air density (kg m$^{-3}$), $a$ is adjustment factor (-). | This is to calculate soil thermal conductivity for frozen and unfrozen conditions (Ballard and Arp, 2005) |
| A12 | $$R_f = e^{c_f T_s}C_{md} + (1 - C_{md})$$ where $c_f$ is soil surface frost adjustment coefficient ($^{o}$C$^{-1}$), $C_{md}$ is maximum | This is used to adjustment thermal |



| | | |
|---|---|---|
| | frost damping coefficient (-), $T_s$ is surface soil temperature ($^{\circ}$C) | conductivity of frozen soil |
| A13 | $$q_{\text{freeze}} = \alpha_h \Delta z \frac{T}{L_f}$$ where $\alpha_h$ is heat transfer coefficient (W m$^{-1}$ $^{\circ}$C$^{-1}$), $\Delta z$ is thickness of soil layer (m), $T$ is soil temperature ($^{\circ}$C), $L_f$ is the latent heat of freezing (J m$^{-3}$). | Redistribution of infiltrating water from high flow to low flow domain |
| | *Energy balance processes* | |
| A14 | $$e_{\text{surf}} = e_s(T_s)e^{\left(\frac{\psi M_{\text{water}} g e_{\text{corr}}}{R(T_s+273.15)}\right)}$$ $$e_{\text{corr}} = 10^{(-\delta_{\text{surf}}\psi_{eg})}$$ where $e_s$ is the vapor pressure (m) at saturation at soil surface temperature $T_s$ ($^{\circ}$C), $\psi$ is the soil water tension (m) and $g$ is the gravitational constant (g m$^{-2}$ s$^{-1}$), $R$ is the gas constant (J $^{\circ}$C$^{-1}$ mol$^{-1}$), $M_{water}$ is the molar mass of water (18 g mol$^{-1}$) and $e_{\text{corr}}$ is the empirical correction factor, $\psi_{eg}$ is a parameter and $\delta_{\text{surf}}$ is a calculated mass balance at the soil surface (m), which is allowed to vary between the parameters $s_{\text{def}}$ and $s_{\text{excess}}$ given in m of water. | Vapor pressure at the soil surface |
| A15 | $$\delta_{\text{surf}}(t) = \max\left(s_{\text{def}}, \min\left(\begin{matrix} s_{\text{excess}}, \delta_{\text{surf}}(t-1)+W_{\text{pool}}+ \\ (q_{in}-E_s-q_{v,s}+i_{\text{drip}}(z_1))\Delta t \end{matrix}\right)\right)$$ where $W_{\text{pool}}$ is the surface water pool (m), $q_{in}$ is the infiltration rate (m s$^{-1}$), $E_s$ is the evaporation rate (m s$^{-1}$), $i_{drip}$ is drip irrigation rate (m s$^{-1}$), $q_{v,s}$, is the vapor flow from soil surface to the central point of the uppermost soil layer (m s$^{-1}$), $s_{\text{def}}$ is maximal surface water deficit (m) and $s_{\text{excess}}$ is maximal surface water excess (m). | Mass balance check at the soil surface |
| A16 | $$r_{aa} = \frac{1}{k^2 u} \ln\left(\frac{z_{\text{ref}}}{z_{\text{OM}}}\right) \ln\left(\frac{z_{\text{ref}}}{z_{\text{OH}}}\right) f(R_{ib})$$ where $u$ is the wind speed (m s$^{-1}$) at the reference height, $z_{\text{ref}}$ (m), $R_{ib}$ is the bulk Richardson number, $k$ is the von Karman constant and $z_{\text{OM}}$ and $z_{\text{OH}}$ are the surface roughness lengths for momentum and heat, respectively (m). | Aerodynamic resistance at stable atmosphere |
| A17 | $$kB^{-1} = \ln\left(\frac{z_{\text{OM}}}{z_{\text{OH}}}\right)$$ where $kB^{-1}$ is a parameter with a default value 0 (implies $z_{\text{OH}} = z_{\text{OM}}$). | Calculation of of $z_{OH}$ from $kB^{-1}$ |
| A18 | $$r_{aa} = \left(\frac{1}{r_{aa}} + r_{a,max}^{-1}\right)^{-1}$$ where $r_{a,\max}^{-1}$ is a parameter for a upper limit of the aerodynamic resistance in extreme stable conditions. | Aerodynamic resistance in extreme stable conditions |
| A19 | $$E_s = \max\left(-e_{\text{max,cond}}, L_v E_s / L_v\right) f_{\text{bare}}$$ where $e_{\text{max,cond}}$ is maximum condensation rate (m s$^{-1}$) for upmost soil layer to maintain water balance, $f_{bare}$ is bare soil fraction. | Soil evaporation limiting factor |





| A20 | $$\varepsilon_{a,Konzelmann} = \left( r_{k1} + r_{k2}\frac{e_a}{T_a+273.15} \right)^{1/4} \left(1-n_c^3\right) + r_{k3}n_c^3$$ <br> where $e_a$ is vapor pressure (m), $T_a$ is air temperature (°C) $n_c$ is cloud fraction; $r_{k1}$, $r_{k2}$, and $r_{k3}$ are parameters. | Longwave radiation estimation |
|---|---|---|
| A21 | $$k_{snow} = s_k \rho_{snow}^2$$ <br> where $s_k$ is empirical parameter (W m$^5$ °C$^{-1}$ kg$^{-2}$), $\rho_{snow}$ is density of snow (kg m$^{-3}$). | Thermal conductivity of snow |
| A22 | $$a_{soil} = a_{dry} + e^{-k_a^{10}\lg\psi}\left(a_{wet}-a_{dry}\right)$$ <br> where $a_{dry}$ is albedo of dry soil, $a_{wet}$ is albedo of wet soil, and $k_a$ is a transform coefficient from wet to dry soil. | Albedo of bare soil |
| A23 | $$a_{snow} = a_{min} + a_1 e^{a_2 S_{age} + a_3 \sum T_a}$$ <br> where $a_{min}$ is minimum albedo of snow, $S_{age}$ is snow age (d), $a_1$, $a_2$, and $a_3$ are parameter. | Albedo of snow |


