# Peer review of "Implementation of salt-induced freezing point depression function into"

_Hydrology and Earth System Sciences, 2018_

## Referee Comment (RC1) · Anonymous Referee #1 · 27 Feb 2019

Thank you for giving me the opportunity to review "Implementation of salt-induced freezing point depression function into CoupModel_v5 for improvement of modelling seasonally frozen soils" by Wu et al.

General comments

The consideration of salt concentrations on the freezing/thawing dynamics of agricultural soils is an important and interesting addition to the CoupModel. I appreciate the study and found it reasonably well written overall. The description of the model and the application of the model are good, however, the introduction lacks a proper literature review on hydrological models which simulate soil-surface water-groundwater flow

under consideration of winter hydrological processes. This is an important aspect to discuss in the introduction, because it allows readers to better understand the novelty of the extended CoupModel approach. Rather than the CoupModel, which only considers a soil column, integrated flow models have the capability of considering entire agricultural fields or watersheds, including freezing/thawing of soils. However, while these integrated flow models (i.e., MIKE SHE, HSPF, SUTRA, HydroGeoSphere, or ParFlow) have many advantages, they are also limited in that they are less rigorous in their mathematical implementations of freezing/thawing processes. This needs to be highlighted in the introduction, with proper referencing of these existing but with regard to winter hydrological processes less rigorous approaches. I provided references to these four modelling codes in my detailed review below.

Another problem of the study is the fact that results and discussion are combined into one section. This is not per se a problem, but as so many analyses were carried out, it is extremely cumbersome to find the more widely applicable information within your results and discussion section. A lot of information gets lost between the extremely site specific performance criteria, parameter sensitivities etc. This is a pity, because the analyses are all very interesting and the findings that derive from the site specific numbers are more widely applicable. Please split the results and discussion section in order to improve the impact of your article. Moreover, include many of the statements which are now placed at the end of each sub-section oin the results and discussion in the conclusions instead. For example statements such as 'In the future development of the CoupModel, we recommend inclusion of a new scheme for water and energy balance on ice coverage.'. This sentence does not belong in the results section, it should form part of the conclusions.

I support the publication of this study in HESS but I suggest minor to moderate revisions as outlined below prior to acceptance for publication. I also strongly advice the authors to consider a professional English editing service, as there are still many small mistakes in the text. I highlight quite a few of those, but I could not pick out every missing 'the'

and so on. For the article to be ready for publication in a high quality journal such as HESS I suggest improving the English with professional help.

Specific comments and technical corrections

Title: remove '_v5' from the title

Abstract: Line 14-15: Change to 'Salt in agricultural fields impacts soil freezing/thawing characteristics and, therefore, soil hydrologic processes' Line 21: Why 'Nevertheless'? I don't understand the logic of this sentence. This is not a contradiction to the previous sentence and doesn't require nevertheless. Line 23: 'However' is not needed Line 26: Change to 'provides' Line 27: Change to 'highlights'

1. Introduction: The introduction is well written overall and the objectives of the study are clearly stated. There is a lack of literature review for already existing models which are capable of simulating freezing/thawing of soils, and the differences between the new approach and these models is not properly discussed. I want to see a better discussion of existing models with proper referencing as well as a discussion on how CoupModel is different or 'better', as the authors claim. Below are some minor corrections as well as 4 other model examples which must be discussed. These are all minor changes, however.

Line 31: Change to 'recognized for their importance' Line 46: Change to 'on the freezing rate of soil' Line 48: Change to 'in the two same agricultural fields as used in this study' Line 55-62: Some references to more advanced numerical models using soil freezing thawing must be included in this paragraph. You only cite simple hydrological models, but not the advanced, process-based flow simulators that are often used to simulate winter hydrological processes. Include at least the following four with proper referencing:

MIKE SHE: Graham, D. N., & Butts, M. B. (2005). Flexible, integrated watershed modelling with MIKE SHE. In V. P. Singh & D. K. Frevert (Eds.), Watershed Models (pp.

245–272). Boca Raton, FL, USA: Taylor and Francis.

HSPF: Bicknell, B. R., Imhoff, J. C., Kittle, J. L. J., Donigian, A. S. J., & Johanson, R. C. (1997). Hydrological Simulation Program-Fortran, User's manual for version 11. Retrieved from Athens, GA, USA.

HydroGeoSphere: Schilling, O. S., Park, Y.-J., Therrien, R., & Nagare, R. M. (2019). Integrated Surface and Subsurface Hydrological Modeling with Snowmelt and Pore Water Freeze-Thaw. Groundwater, 57(1), 63-74. doi:10.1111/gwat.12841

SUTRA: Voss, C. I., & Provost, A. (2010). SUTRA: A model for saturated-unsaturated variable density groundwater flow with solute or energy transport. Version 2.2. Retrieved from Reston, VA.

Line 63: Change to 'However, there are large uncertainties...' Line 64-65: Change to '...uncertainty analysis methods are often applied by combining ....' Line 66: Change to 'is one of the commonly applied methods for uncertainty analysis...' Line 69: Change to 'in simulations, a concept known as equifinality (Beven 2006)' Line 70: Remove the last sentence on line 70. If you did this in your study, then describe it in the methods section. In the introduction this information is not needed. Line 78: remove 'grown' Line 86-87: change to 'Thus, we extended CoupModel to consider impacts of...' Line 88-89: change to ' The main objective was to 1) extend CoupModel by considering effects of salt on the freezing point; 2) identify parameter sensitivity; 3) analyze the uncertainty in modeling soil hydrology in seasonally frozen agricultural soils.'

2. Material and Methods Lines 97-99: You suggest that the typical clay type soils of Qianguo are low porosity. However, you give a bulk porosity of 0.46, which is massive! Discuss this better or rethink your statements.

Line 99-100: the water table fluctuates between 1.5 and 2m. Is that the variation in GW levels or is the variation 0.5m and the 1.5-2m is the depth at which the water table is normally situated? not clear.

Line 102: change 'from' to 'form'

Figure 1: change the color of the blue dots in the figure to a brighter color, it's almost impossible to see them. The figure is overall very complicated. Can you please rearrange the subfigures in a better way?

Line 109: Exact dates are provided for Yonglian, but not for Qianguo. Line 111-112: What is the soil type? Porosity is the same as in the first site. Again, is this the water table fluctuation in variation or the depth at which the water table normally is?

Line 115-116: reference Figure 1, so that readers know that the selected study plots are illustrated in Figure 1

Table 1: Site: The abbreviations NE and IM have not yet been introduced

Line 127: what do you mean by 'the daily temperature data were collected? Did you measure just once a day? Is that value representative of the average daily temperature? Or did you record at a higher interval and then averaged to daily average temperatures?

Line 149: delete '(by hearing the voice)' Line 153: Specify which meteorological stations, and provide the data alongside all other data. Or state where the data can be obtained. All data used for the study must be available or made available.

3. CoupModel

Line 157: remove '_v5' and change the section title to 'Extension of the CoupModel' Line 158: change to 'The model domain covered the top 6m of the soil, with unit area considered' Line 163: change to 'in this study are' Section 3.1.: Change all to present tense. Line 166: change to 'CoupModel solves the coupled...'

Line 381: change to 'for analysis of water, energy'...

4. Results and Discussion Here you combine Results and Discussion into one single section. While some strictly dismiss this combination, where the circumstances allow

it I don't mind combining both in one section. However, your Results and Discussion is very long with a lot of Figures, aspects, numbers etc. The discussion of the different important findings gets lost among the many numbers, performance criteria, sensitivities etc. Therefore, I suggest renaming this section 'Results' and adding a separate section 'Discussion' where you summarize the findings of the different analyses, i.e., where you state clearly and without using too many numbers which parameters were sensitive due to what reason etc.

Line 410: change to' which is used to estimate' Figure 5: This is a very poorly described figure. It isn't at all clear which plot relates to which parameter. Which parameters are sensitive to what outcome?? Explain better!! The same is true for Figure 6 !! Line 447: change to 'Even though we have...'

5. Conclusions The conclusions lack precise statements for what is still needed. In the results and discussion section, things like 'In the future development of the CoupModel, we recommended inclusion of a new scheme for water and energy balance on ice coverage.' (lines 492-493). This information and all the other statements on what is still required must be provided in the conclusions. This comment is similar to my comment above: Because you combined results and discussion into one single section all this information gets lost! You carried out so many interesting analyses with your extended CoupModel that a combined results and discussion section is not appropriate. Please consider splitting results, discussion and conclusions so that the readers can more easily find the important findings and conclusions, rather than having to go through all the extremely site specific performance and sensitivity numbers! The specific numbers are have no widely applicable meaning, but the conclusions you draw from them do. Therefore, make them more visible to increase the impact of your paper.

Line 543: change to 'are coupled in agricultural fields'

---

## Referee Comment (RC2) · Anonymous Referee #2 · 19 Mar 2019

Summary: The authors have presented a study in which they modify the CoupModel to consider the effects of dissolved solutes on the soil-water freezing point depression. The study is, for the most part, a valuable contribution to the field of cold regions hydrological modeling. However, the structure of the manuscript makes it hard to follow in many places. A clearer description of the model processes is needed, and the results and discussion need to be split into two separate sections. There are also many small grammatical errors throughout, I have highlighted some but there were too many for me to individually address.

I also have somewhat of an issue with the entire approach to this kind of modeling. The

authors incorporate a new process in the model, but rather than systematically testing the impact of the new physics on the model outputs against simple test cases that isolate the effect of one process on another, they authors try to simulate a field experiment with a multitude of different and interacting processes. An alternative approach would have been to simulate a much simpler, highly controlled experiment (for e.g. the authors could have simulated the soil column experiment of Stähli and Stadler (1997)), to test the performance of the newly implemented freezing-point depression relation. After that the authors could have applied their model to their field site. However, I admit this is a philosophical difference of opinion and there are other researchers who would advocate for this approach. Thus, I only bring up this point for the consideration of the associate editor. I believe the content of the article fits well within the scope of HESS and could be useful for the targeted audience. However, in my opinion, the manuscript could be suitable for publication in HESS only after major revisions are done to the structure and quality of writing of the manuscript.

General comments:

Lines 131 – 135: Did this pre-calibrated relationship consider the in-situ soil salinity already present and its effect in the electrical potential of the soil water? If the soils at both sites are affected by salinization issues, then wouldn't in-situ salinity be an important consideration? More discussion of this is warranted. Furthermore, was any in-situ salinity measured? This could potentially a major issue in the field data collection. Can the authors give reason as to why this was not attempted? Possibly due to equipment trouble?

Lines 158 – 164: Was a sensitivity test carried out to ensure model discretization and time-stepping choices did not affect the numerical solution? In my experience, when modeling water partitioning at the soil surface, node spacing may sometimes have to be smaller than 10 cm.

Lines 258 – 260: Why was initial salt concentration, precipitation salt concentration and

irrigation salt concentration used calibration parameters? Couldn't these numbers be obtained from the site measurements? I have an issue with the initial salt concentration being used a calibration parameter, it will obviously influence the tracer transport. This brings me back to my earlier point amount measuring the in-situ salinity of the soil. It is also possible to estimate soil salinity using TDR (e.g. Stähli and Stadler, 1997). One could also envision the initial salt concentration profile being estimated from the change in electrical conductivity after the initial application of the Br- tracer.

Section 3: This section in general is written in a rather convoluted manner and is not the easiest for the reader to understand. In general, I think the entire section 3 needs to be rewritten to address the many grammatical issues and lack of clarity in the description of the processes and model calibration/validation approach.

Section 3: I think the authors need to more clearly emphasize that they employ both Esbey's (1992) bypass flow routine as well as Stahli et al.'s (1996) dual-domain hydraulic conductivity concept to simulate water flow in frozen soil, i.e. you have high and low flow domains in the soil matrix and additional by-pass flow through macropores. The authors need to careful to clearly identify the different processes being simulated as there is overlap in the terminology used to described different processes in the model. For example, in lines 182 – 184 you use the term high-flow domain to describe the bypass routine's ability to route water directly in the underlying soil layer when the infiltration capacity of the soil matrix is reached. However, you also you the term high-flow domain in lines 196 – 199 to describe the frozen soil hydraulic conductivity model of Stahli et al. (1996). Thus, you need to be very clear about which frozen soil flow process you are describing. It must also be noted that this approach, while rather complex, still has its conceptual limitations as refreezing of infiltrated water can also occur in macropores (Watanabe and Kugisaki, 2017). While the concept of Stähli et al. (1996) has widely been used to incorporate the effects of preferential flow and refreezing of infiltrated water, you also employ a by-pass flow routine which does not consider the effect of refreezing along macropores. Some discussion of this is warranted. See the recent review of Mohammed et al. (2018) on macropore flow in frozen soils.

Section 4: This section needs to be split into two separate Results and Discussion sections.

Line 447: The authors did not develop a new relationship, other frozen soil models (e.g. SHAW - Flerchinger Saxton (1989)) have long incorporated the relationship between osmotic potential and soil freezing temperature.

Lines 452 – 453: Again, it is not enough to simply state that they impact soil heat and water transport. How specifically does it impact heat and water transport?

Lines 511 – 513: More discussion the model misfit at lower depths is needed. From figure 8, at one site the model overestimates soil water at depth and at the other site it underestimates it. What are the reasons for this? Is it an improper representation of the vadose zone flow processes (as the authors include quite a complicated representation of the soil water flow processes) or a misrepresentation of the lower boundary condition and influence of groundwater, or some combination of both?

Lines 527 – 538: More discussion of the large difference in the model and measured salt storage is needed. What was the mass recovery of the applied Br- tracer relative to the application? This would give some insight into the flow processes affecting solute transport. For example, very little BR- was measured at the Qianguo site... was this due to leaving due to preferential flow to groundwater or increased retention of the tracer near the surface? This could also be a possible reason for your mismatch of soil water storage at deeper depths (see comment above).

Additional references: Espeby, B. 1992. Coupled simulations of water flow from a field-investigated glacial till slope using a quasi-two-dimensional water and heat model with bypass flow. Journal of Hydrology 131:105–132.

Flerchinger, G.N., and K.E. Saxton. 1989. Simultaneous heat and water model of a

freezing snow residue–soil system: 1. Theory and development. Transactions of the ASAE 32:565–571.

Mohammed AA, Kurylyk BL, Cey EE, Hayashi M. 2018. Snowmelt infiltration and macropore flow in frozen soils: overview, knowledge gaps, and a conceptual framework. Vadose Zone Journal 17(1).

Stähli, M., P. Jansson, and L.C. Lundin. 1996. Preferential water flow in a frozen soil: A two-domain model approach. Hydrological Processes 10:1305–1316.

Stähli M, Stadler D. 1997. Measurement of water and solute dynamics in freezing soil columns with time domain reflectometry. Journal of Hydrology 195(1-4):352-369.

Watanabe, K., and Y. Kugisaki. 2017. Effect of macropores on soil freezing and thawing with infiltration. Hydrological Processes 31:270–278.

Technical corrections:

Abstract:

Line 15 – 16: Delete 'In this context' from the sentence, unnecessary.

Line 17: '. . . influences of salt on cold region hydrology' is too vague a statement. Reword to be more specific, for example: '. . . influences of soil salinity on soil water and heat transport'.

Line 18: Modify sentence to 'We modified the CoupModel to simulate the impacts of salinity on soil freezing point depression'.

Line 21: Delete words 'into CoupModel', unnecessary.

Line 26: Change 'provided' to 'provides'.

Introduction:

Line 35: Awkwardly worded sentence, change to something along the lines of 'Knowledge on soil freezing and thawing is needed to better understand mechanisms. . .'.

Line 48: Change to '... in the two same agricultural fields in this study...'

Line 61: Should be '... agricultural fields'.

Line 62: Modify to '... and other cold region ecosystems'

Line 63: Should be 'However there are large uncertainties...'

Line 64: Modify to 'and coupled transport processes'.

Line 64 – 65: Modify sentence to '... uncertainty analysis methods have been utilized by...'.

Line 67: modify to '... is a commonly used...'.

Line 70: Modify to 'GLUE is performed...'.

Line 75: Delete '... in the northern part of China.' Redundant.

Line 86 – 87: Modify sentence to 'We modified the CoupModel to consider the impacts of salinity on soil freezing...'.

Line 89: Modify to '... 2) perform a sensitivity analysis on the new model'.

Line 90: Modify to '... in modeling hydrological process in seasonally frozen soils.'.

Material and Methods:

Line 93: Should be '... in northern China.'

Line 95: Should be 'Field experiments at...'.

Line 128: Should be 'During the soil freeze-thaw period at...'.

Line 192: you not describe what the pF value is.

Line 229: Change to 'latent heat transfer:'

Line 257: Lateral boundaries? Isn't Coup a 1-D model? Why would it need a lateral

boundary condition?

Line 285: You refer to equation (5), but are talking about the surface energy balance, I think you meant equation (15).

Line 291: need a citation for the Richardson equation as readers may not be familiar with the relationship.

Line 323: I think you mean H is the total sensible heat stored in the soil, not total energy?

Results and Discussion:

Figure 3: Figure 3a should be modified to use solid circles like Figure 3b. It would make the Fig. 3a easier to read.

Conclusions:

Line 543: Should be '. . . are coupled in agricultural fields'.

Line 566: Replace 'would be very necessary in investigation' to 'is still needed to improve understanding of'.

Please also note the supplement to this comment:
https://www.hydrol-earth-syst-sci-discuss.net/hess-2018-466/hess-2018-466-RC2-supplement.pdf

---

## Author Comment (AC1) · 25 Apr 2019

We thank the reviewer for the useful comments and kind suggestions to improve this manuscript. We have carefully considered them all and revised the manuscript accordingly. In the following, we have provided point-by-point responses (red) to the reviewer's comments (black).

**Reviewer #1**

Thank you for giving me the opportunity to review "Implementation of salt-induced freezing point depression function into CoupModel_v5 for improvement of modelling seasonally frozen soils" by Wu et al.

General comments

The consideration of salt concentrations on the freezing/thawing dynamics of agricultural soils is an important and interesting addition to the CoupModel. I appreciate the study and found it reasonably well written overall. The description of the model and the application of the model are good, however, the introduction lacks a proper literature review on hydrological models which simulate soil-surface water-groundwater flow under consideration of winter hydrological processes. This is an important aspect to discuss in the introduction, because it allows readers to better understand the novelty of the extended CoupModel approach. Rather than the CoupModel, which only considers a soil column, integrated flow models have the capability of considering entire agricultural fields or watersheds, including freezing/thawing of soils. However, while these integrated flow models (i.e., MIKE SHE, HSPF, SUTRA, HydroGeoSphere, or ParFlow) have many advantages, they are also limited in that they are less rigorous in their mathematical implementations of freezing/thawing processes. This needs to be highlighted in the introduction, with proper referencing of these existing but with regard to winter hydrological processes less rigorous approaches. I provided references to these four modelling codes in my detailed review below.

**Response to general comments**: We agreed with the reviewer. This is a good point. We have mentioned those models the reviewer has suggested and highlighted that the CoupModel has more rigorous mathematical implantations than those models.

Another problem of the study is the fact that results and discussion are combined into one section. This is not per se a problem, but as so many analyses were carried out, it is extremely cumbersome to find the more widely applicable information within your results and discussion section. A lot of information gets lost between the extremely site specific performance criteria, parameter sensitivities etc. This is a pity, because the analyses are all very interesting and the findings that derive from the site specific numbers are more widely applicable. Please split the results and discussion section in order to improve the impact of your article. Moreover, include many of the statements which are now placed at the end of each sub-section in the results and discussion in the conclusions instead. For example statements such as 'In the future development of the CoupModel, we recommend inclusion of a new scheme for water and energy balance on ice coverage.'. This sentence does not belong in the results section, it should form part of the conclusions.

*Response to comment*: We agreed with the reviewer and have split the section "Results and Discussion" into two separate sections.

I support the publication of this study in HESS but I suggest minor to moderate revisions as outlined below prior to acceptance for publication. I also strongly advice the authors to consider a professional English editing service, as there are still many small mistakes in the text. I highlight quite a few of those, but I could not pick out every missing 'the' and so on. For the article to be ready for publication in a high quality journal such as HESS I suggest improving the English with professional help.

*Response to comment*: Thanks for the suggestion on improving English. We have asked a professional English editing service to help us improve language of the manuscript.

Specific comments and technical corrections

Title: remove '_v5' from the title

*Response to comment*: We have changed the title as "Modelling seasonal freezing and thawing of agriculture soils in northern China: Implementing a salt-induced freezing point depression function in CoupModel".

Abstract: Line 14-15: Change to 'Salt in agricultural fields impacts soil freezing/thawing characteristics and, therefore, soil hydrologic processes' Line 21: Why 'Nevertheless'? I don't understand the logic of this sentence. This is not a contradiction to the previous sentence and doesn't require nevertheless. Line 23: 'However' is not needed Line 26: Change to 'provides' Line 27: Change to 'highlights'

*Response to comment*: We have added the suggested sentence. In Line 21 and 23, "Nevertheless" and "However" have been deleted. In Line: 27, we have changed these words as suggested.

Introduction: The introduction is well written overall and the objectives of the study are clearly stated. There is a lack of literature review for already existing models which are capable of simulating freezing/thawing of soils, and the differences between the new approach and these models is not properly discussed. I want to see a better discussion of existing models with proper referencing as well as a discussion on how CoupModel is different or 'better', as the authors claim. Below are some minor corrections as well as 4 other model examples which must be discussed. These are all minor changes, however.

*Response to comment*: We have mentioned those suggested models in Introduction and added a discussion on how CoupModel is different from them.

Line 31: Change to 'recognized for their importance' Line 46: Change to 'on the freezing rate of soil' Line 48: Change to 'in the two same agricultural fields as used in this study'

*Response to comment*: We have made the suggested changes.

Line 55-62: Some references to more advanced numerical models using soil freezing thawing must be included in this paragraph. You only cite simple hydrological models, but not the advanced, process-based flow simulators that are often used to simulate winter hydrological processes. Include at least the following four with proper referencing:

MIKE SHE: Graham, D. N., & Butts, M. B. (2005). Flexible, integrated watershed modelling with MIKE SHE. In V. P. Singh & D. K. Frevert (Eds.), Watershed Models (pp. 245–272). Boca Raton, FL, USA: Taylor and Francis.

HSPF: Bicknell, B. R., Imhoff, J. C., Kittle, J. L. J., Donigian, A. S. J., & Johanson, R. C. (1997). Hydrological Simulation Program-Fortran, User's manual for version 11. Retrieved from Athens, GA, USA.

HydroGeoSphere: Schilling, O. S., Park, Y.-J., Therrien, R., & Nagare, R. M. (2019). Integrated Surface and Subsurface Hydrological Modeling with Snowmelt and Pore Water Freeze-Thaw. Groundwater, 57(1), 63-74. doi:10.1111/gwat.12841

SUTRA: Voss, C. I., & Provost, A. (2010). SUTRA: A model for saturated-unsaturated variable density groundwater flow with solute or energy transport. Version 2.2. Retrieved from Reston, VA.

*Response to comment*: We have added the suggested models in Introduction and discussed them.

Line 63: Change to 'However, there are large uncertainties...' Line 64-65: Change to '...uncertainty analysis methods are often applied by combining ....' Line 66: Change to 'is one of the commonly applied methods for uncertainty analysis...' Line 69: Change to 'in simulations, a concept known as equifinality (Beven 2006)' Line 70: Remove the last sentence on line 70. If you did this in your study, then describe it in the methods section. In the introduction this information is not needed. Line 78: remove 'grown' Line 86-87: change to 'Thus, we extended CoupModel to consider impacts of...' Line 88-89: change to ' The main objective was to 1) extend CoupModel by considering effects of salt on the freezing point; 2) identify parameter sensitivity; 3) analyze the uncertainty in modeling soil hydrology in seasonally frozen agricultural soils.'

*Response to comment*: We have made the suggested changes.

2. Material and Methods Lines 97-99: You suggest that the typical clay type soils of Qianguo are low porosity. However, you give a bulk porosity of 0.46, which is massive!

Discuss this better or rethink your statements.

*Response to comment*: We have changed the sentences as "The soil profile at Site Qianguo is homogeneous, with porosity of 0.46, bulk density of 1.42 g cm$^{-3}$ and soil texture characterized as clay loam (Table 1)".

Line 99-100: the water table fluctuates between 1.5 and 2m. Is that the variation in GW levels or is the variation 0.5m and the 1.5-2m is the depth at which the water table is normally situated? not clear.

*Response to comment*: "1.5 to 2.0 m" refers to the groundwater table depth below soil surface. We have clarified it in Line 99-100.

Line 102: change 'from' to 'form'

*Response to comment*: We have changed it in Line 102.

Figure 1: change the color of the blue dots in the figure to a brighter color, it's almost impossible to see them. The figure is overall very complicated. Can you please rearrange the subfigures in a better way?

*Response to comment*: Thanks for the suggestion. We have made the following modification for Fig 1. Firstly, we have changed the blue dots to the green dots to make them easier to read. Secondly, we divided this figure into three sub-figures and used a), b) and c) to make them easier for reading. The caption have been revised accordingly.

Line 109: Exact dates are provided for Yonglian, but not for Qianguo. Line 111-112: What is the soil type? Porosity is the same as in the first site. Again, is this the water table fluctuation in variation or the depth at which the water table normally is?

*Response to comment*: We have provided the dates for Qianguo, and the soil type as well. We have also explained the groundwater fluctuation referred to the groundwater table depth below soil surface.

Line 115-116: reference Figure 1, so that readers know that the selected study plots are illustrated in Figure 1

***Response to comment***: We have added Figure 1 as a reference to the text.

Table 1: Site: The abbreviations NE and IM have not yet been introduced

***Response to comment***: We have corrected the site name in Table 1, as Qianguo and Yonglian.

Line 127: what do you mean by 'the daily temperature data were collected? Did you measure just once a day? Is that value representative of the average daily temperature? Or did you record at a higher interval and then averaged to daily average temperatures?

***Response to comment***: Temperature is measured every day at noon. The reason for measuring the temperature once a day is that the limited resource doesn't allow us to install a data-logger in a remote field. We have clarified it in Line 127.

Line 149: delete '(by hearing the voice)' Line 153: Specify which meteorological stations, and provide the data alongside all other data. Or state where the data can be obtained. All data used for the study must be available or made available.

***Response to comment***: we have deleted '(by hearing the voice)' in Line 149. The meteorological stations are installed at each site in the field and run by local water management unit. The data are available upon request to the correspondence author. We have addressed it in Line 153.

3. CoupModel

Line 157: remove '_v5' and change the section title to 'Extension of the CoupModel'

***Response to comment***: We have changed the section title, in Line 157.

Line 158: change to 'The model domain covered the top 6m of the soil, with unit area considered' Line 163: change to 'in this study are' Section 3.1.: Change all to present tense. Line 166: change to 'CoupModel solves the coupled...'

***Response to comment***: We have changed the expression in Line 158, 163, 166, and changed all to present tense in model description.

Line 381: change to 'for analysis of water, energy'...

***Response to comment***: We have changed it in Line 381.

4. Results and Discussion Here you combine Results and Discussion into one single section. While some strictly dismiss this combination, where the circumstances allow it I don't mind combining both in one section. However, your Results and Discussion is very long with a lot of Figures, aspects, numbers etc. The discussion of the different important findings gets lost among the many numbers, performance criteria, sensitivities etc. Therefore, I suggest renaming this section 'Results' and adding a separate section 'Discussion' where you summarize the findings of the different analyses, i.e., where you state clearly and without using too many numbers which parameters were sensitive due to what reason etc.

***Response to comment***: Thanks for the suggestion. We have separated Results from Discussion. In Results, we mainly showed the results in modeling. In Discussion we have discussed parameter uncertainties, model performance, and simulation results uncertainties etc.

Line 410: change to' which is used to estimate' Figure 5: This is a very poorly described figure. It isn't at all clear which plot relates to which parameter. Which parameters are sensitive to what outcome?? Explain better!! The same is true for Figure 6 !! Line 447: change to 'Even though we have...'

***Response to comment***: We are sorry for the poorly described figures. We have improved the figures by using larger fonts for the important parameters in the sub-figures and to make the figures easier to read. Accordingly, we have revised the related section (Section 4.2).

5. Conclusions The conclusions lack precise statements for what is still needed. In the results and discussion section, things like 'In the future development of the CoupModel, we recommended inclusion of a new scheme for water and energy balance on ice coverage.' (lines 492-493). This information and all the other statements on what is still required must be provided in the conclusions. This comment is similar to my comment above: Because you combined results and discussion into one single section all this information gets lost! You carried out so many interesting analyses with your extended CoupModel that a combined results and discussion section is not appropriate. Please consider splitting results, discussion and conclusions so that the readers can more easily find the important findings and conclusions, rather than having to go through all the extremely site specific performance and sensitivity numbers! The specific numbers are have no widely applicable meaning, but the conclusions you draw from them do. Therefore, make them more visible to increase the impact of your paper.

***Response to comment***: Thanks for the suggestions on improving our paper. We have revised the Conclusion part by emphasizing the needs in the future.

Line 543: change to 'are coupled in agricultural fields'

***Response to comment***: We have changed it in Line 543.

---

## Author Comment (AC2) · 25 Apr 2019

We thank the reviewer for the comments and kind suggestions to improve this manuscript. We have carefully considered them all and revised the manuscript accordingly. In the following, we have provided point-by-point responses (red) to the reviewer's comments (black).

**Reviewer #2**

Summary:

The authors have presented a study in which they modify the CoupModel to consider the effects of dissolved solutes on the soil-water freezing point depression. The study is, for the most part, a valuable contribution to the field of cold regions hydrological modeling. However, the structure of the manuscript makes it hard to follow in many places. A clearer description of the model processes is needed, and the results and discussion need to be split into two separate sections. There are also many small grammatical errors throughout, I have highlighted some but there were too many for me to individually address.

*Response to comment*: We have reorganized the structure of the manuscript and split the section "Result and Discussion" into two separate sections.

I also have somewhat of an issue with the entire approach to this kind of modeling. The authors incorporate a new process in the model, but rather than systematically testing the impact of the new physics on the model outputs against simple test cases that isolate the effect of one process on another, they authors try to simulate a field experiment with a multitude of different and interacting processes. An alternative approach would have been to simulate a much simpler, highly controlled experiment (for e.g. the authors could have simulated the soil column experiment of Stähli and Stadler (1997)), to test the performance of the newly implemented freezing-point depression relation. After that the authors could have applied their model to their field site. However, I admit this is a philosophical difference of opinion and there are other researchers who would advocate for this approach. Thus, I only bring up this point for the consideration of the associate editor. I believe the content of the article fits well within the scope of HESS and could be useful for the targeted audience. However, in my opinion, the manuscript could be suitable for publication in HESS only after major revisions are done to the structure and quality of writing of the manuscript.

*Response to comment*: Yes, we agreed with the reviewer that the soil column experiment, for instance, similar to Stähli and Stadler, (1997) may be a simpler and straightforward approach to test the freezing-point depression curve in relation to the salt concentration, however, experimental studies often encounter limitations, like shorter periods of implementation, and making assumptions of simpler boundary conditions, which is contradictory to our field conditions, where the processes related to snow, soil evaporation, and groundwater table depth and their interactions with soil heat and water transfer provide more complicated upper and bottom boundary conditions. Moreover, the advantages of our approach (a Monte-Carlo based calibration) not only ensure parameterizations are efficient to describe the observations but also identify the relative importance of parameters to the model performance.

Lines 131 – 135: Did this pre-calibrated relationship consider the in-situ soil salinity already present and its effect in the electrical potential of the soil water? If the soils at both sites are affected by salinization issues, then wouldn't in-situ salinity be an important consideration? More discussion of this is warranted. Furthermore, was any in-situ salinity measured? This could potentially a major issue in the field data collection. Can the authors give reason as to why this was not attempted? Possibly due to equipment trouble?

*Response to comment*: We have made the pre-calibration in the laboratory by using soil samples with different water contents setups. Yes, when we use this pre-calibrated relationship in the field, it will result in some uncertainties since in the field the salinity of soil will change due to the dynamics of water and salt. This has been discussed in this revision.

As to the measurement of in-situ salinity, we have tried by using some instruments. Unfortunately, the instruments did not work properly even with a pre-calibrated relationship because the salinization of soil made the instruments eroded.

Besides, there is still an issue even if the instrument works in the field. Since we calibrate the instrument in laboratory with unfrozen soil, when we apply it to frozen soil, the changes in liquid water content due to soil solution condensation will greatly influence accuracy of measurements of soil salts in the field. Thus, currently the salt contents measured from the collected soil samples is more reliable than the in-situ measurements we have tried to attain. Nevertheless, it is worth mentioning that we are developing the TDR probes for simultaneously measurements of soil water, temperature and salt and have tested them in the laboratory. Hopefully, we can use them for future in-situ measurements after a proper calibration.

Lines 158 – 164: Was a sensitivity test carried out to ensure model discretization and timestepping choices did not affect the numerical solution? In my experience, when modeling water partitioning at the soil surface, node spacing may sometimes have to be smaller than 10 cm.

*Response to comment*: The size of model discretization and the number of iterations per day (i.e. timestep) is based on previous modelling experience of CoupModel (e.g. Wu et al., 2016, 2018). In this study, our results (Fig. 7) have indicated that the size of model discretization is sufficient to resolve model precision against measurements. The depth of the top soil layer is 10 cm, which corresponds to the measurement at 5 cm depth, assuming that the soil texture within each layer is homogenous.

Lines 258 – 260: Why was initial salt concentration, precipitation salt concentration and irrigation salt concentration used calibration parameters? Couldn't these numbers be obtained from the site measurements? I have an issue with the initial salt concentration being used a calibration parameter, it will obviously influence the tracer transport. This brings me back to my earlier point amount measuring the in-situ salinity of the soil. It is also possible to estimate soil salinity using TDR (e.g. Stähli and Stadler, 1997). One could also envision the initial salt concentration profile being estimated from the change in electrical conductivity after the initial application of the Br- tracer.

*Response to comment*: We used the initial salt concentration, precipitation salt concentration and irrigation salt concentration as calibration parameters since we found there were large differences in the field on the initial salt concentrations when we measured the initial salt profiles at various points of the field. The model is sensitive to the initial salt concentrations. So we set the mean values from different points as the initial values and also put it in the model calibration. For precipitation salt concentration, we did not have measurements, so we just used the model default values. But in the model sensitivity analysis, we found the model is sensitive to this parameter, so we also put it in the model calibration. As to irrigation salt concentration, we measured the irrigation water several times during the irrigation period, and noticed that their values were not stable. Model is also shown sensitive to the irrigation salt concentration. Thus, that is why we used the measured values as prior, and also put them in the model calibration.

We agreed that TDR has shown a promising potential in estimating soil salinity by Stahli and Stadler, 1997. In our case, we have tried it in laboratory. Unfortunately, we failed due to a quite high salinity in the field

soil samples which increased resistance of TDR probes and even made the measurements of soil water unrealistic. Therefore, we decided to adopt the TDR probe in laboratory to accurately measure soil salinity.

Section 3: This section in general is written in a rather convoluted manner and is not the easiest for the reader to understand. In general, I think the entire section 3 needs to be rewritten to address the many grammatical issues and lack of clarity in the description of the processes and model calibration/validation approach.

*Response to comment*: We thank the reviewer to point out the language issue of Section 3. We have rewritten it.

Section 3: I think the authors need to more clearly emphasize that they employ both Esbey's (1992) bypass flow routine as well as Stahli et al.'s (1996) dual-domain hydraulic conductivity concept to simulate water flow in frozen soil, i.e. you have high and low flow domains in the soil matrix and additional by-pass flow through macropores. The authors need to careful to clearly identify the different processes being simulated as there is overlap in the terminology used to described different processes in the model. For example, in lines 182 – 184 you use the term high-flow domain to describe the bypass routine's ability to route water directly in the underlying soil layer when the infiltration capacity of the soil matrix is reached. However, you also you the term high-flow domain in lines 196 – 199 to describe the frozen soil hydraulic conductivity model of Stahli et al. (1996). Thus, you need to be very clear about which frozen soil flow process you are describing. It must also be noted that this approach, while rather complex, still has its conceptual limitations as refreezing of infiltrated water can also occur in macropores (Watanabe and Kugisaki, 2017). While the concept of Stähli et al. (1996) has widely been used to incorporate the effects of preferential flow and refreezing of infiltrated water, you also employ a by-pass flow routine which does not consider the effect of refreezing along macropores. Some discussion of this is warranted. See the recent review of Mohammed et al. (2018) on macropore flow in frozen soils.

*Response to comment*: Thanks for the comment. We find that this is a good point which will be clarified in the revised manuscript. We have read about the three papers suggested here and referred them in our discussions in the revision.

Section 4: This section needs to be split into two separate Results and Discussion sections.

*Response to comment*: We have done it in the revision.

Line 447: The authors did not develop a new relationship, other frozen soil models (e.g. SHAW - Flerchinger Saxton (1989)) have long incorporated the relationship between osmotic potential and soil freezing temperature.

*Response to comment*: We have deleted this sentence.

Lines 452 – 453: Again, it is not enough to simply state that they impact soil heat and water transport. How specifically does it impact heat and water transport?

*Response to comment*: Thanks for the suggestions. We have added some explanations on how salt affects soil temperature and soil water.

Lines 511 – 513: More discussion the model misfit at lower depths is needed. From figure 8, at one site the model overestimates soil water at depth and at the other site it underestimates it. What are the reasons for this? Is it an improper representation of the vadose zone flow processes (as the authors include quite a complicated representation of the soil water flow processes) or a misrepresentation of the lower boundary condition and influence of groundwater, or some combination of both?

*Response to comment*: Thanks for the suggestions. It is really a good point. We have added more discussions on this. Yes, we noticed that in figure 8, soil water was overestimated at Site Qianguo and underestimated at Site Yonglian. For these two sites, their hydrological processes are quite different. At Site Qianguo, the near-surface soil water is more influenced by the snow cover, which means that the infiltration of snowmelt will affect soil water transport. Therefore, as the reviewer suggest, the model bias might be due to an improper representation of boundary conditions and soil infiltration. At Site Yonglian, the major attributor can be the ice cover formed after irrigation. The irrigated water did not fully infiltrate into the soil profile as the model simulates, thus causing an overestimate in the model. This has led to a large influence on the surface water and energy balance in the field. We have discussed these issues in more detail in the revised manuscript.

Lines 527 – 538: More discussion of the large difference in the model and measured salt storage is needed. What was the mass recovery of the applied Br- tracer relative to the application?

This would give some insight into the flow processes affecting solute transport. For example, very little BR- was measured at the Qianguo site… was this due to leaving due to preferential flow to groundwater or increased retention of the tracer near the surface? This could also be a possible reason for your mismatch of soil water storage at deeper depths (see comment above).

*Response to comment*: That's a good point. We have calculated the mass recovery of Br- in another paper (Wang et al., 2016, Soil Science) and found that the recovery was above 80%. We appreciate the insight of reviewer on the flow processes affecting solute transport and thus we have added a discussion on how the preferential flow may explain soil water and salt storage mismatch.

Additional references:

Espeby, B. 1992. Coupled simulations of water flow from a field-investigated glacial till slope using a quasi-two-dimensional water and heat model with bypass flow. Journal of Hydrology 131:105–132.

Flerchinger, G.N., and K.E. Saxton. 1989. Simultaneous heat and water model of a freezing snow residue–soil system: 1. Theory and development. Transactions of the ASAE 32:565–571.

Mohammed AA, Kurylyk BL, Cey EE, Hayashi M. 2018. Snowmelt infiltration and macropore flow in frozen soils: overview, knowledge gaps, and a conceptual framework. Vadose Zone Journal 17(1).

Stähli, M., P. Jansson, and L.C. Lundin. 1996. Preferential water flow in a frozen soil: A twodomain model approach. Hydrological Processes 10:1305–1316.

Stähli M, Stadler D. 1997. Measurement of water and solute dynamics in freezing soil columns with time domain reflectometry. Journal of Hydrology 195(1-4):352-369.

Watanabe, K., and Y. Kugisaki. 2017. Effect of macropores on soil freezing and thawing with infiltration. Hydrological Processes 31:270–278.

Technical corrections:

Abstract:

Line 15 – 16: Delete 'In this context' from the sentence, unnecessary.

*Response to comment*: We have deleted it.

Line 17: '… influences of salt on cold region hydrology' is too vague a statement. Reword to be more specific, for example: '… influences of soil salinity on soil water and heat transport'.

*Response to comment*: We have revised it as the reviewer suggested.

Line 18: Modify sentence to 'We modified the CoupModel to simulate the impacts of salinity on soil freezing point depression'.

*Response to comment*: We have revised it as the reviewer suggested.

Line 21: Delete words 'into CoupModel', unnecessary.

*Response to comment*: We have revised it as the reviewer suggested.

Line 26: Change 'provided' to 'provides'.

*Response to comment*: We have revised it as the reviewer suggested.

Introduction:

Line 35: Awkwardly worded sentence, change to something along the lines of 'Knowledge on soil freezing and thawing is needed to better understand mechanisms…'.

*Response to comment*: We have revised it as the reviewer suggested.

Line 48: Change to '… in the two same agricultural fields in this study…'

*Response to comment*: We have revised it as the reviewer suggested.

Line 61: Should be '… agricultural fields'.

*Response to comment*: We have revised it as the reviewer suggested.

Line 62: Modify to '… and other cold region ecosystems'

*Response to comment*: We have revised it as the reviewer suggested.

Line 63: Should be 'However there are large uncertainties…'

*Response to comment*: We have revised it as the reviewer suggested.

Line 64: Modify to 'and coupled transport processes'.

*Response to comment*: We have revised it as the reviewer suggested.

Line 64 – 65: Modify sentence to '… uncertainty analysis methods have been utilized by…'.

*Response to comment*: We have revised it as the reviewer suggested.

Line 67: modify to '… is a commonly used…'.

*Response to comment*: We have revised it as the reviewer suggested.

Line 70: Modify to 'GLUE is performed…'.

*Response to comment*: We have deleted it.

Line 75: Delete '… in the northern part of China.' Redundant.

*Response to comment*: We have revised it as the reviewer suggested.

Line 86 – 87: Modify sentence to 'We modified the CoupModel to consider the impacts of salinity on soil freezing…'.

*Response to comment*: We have revised it as the reviewer suggested.

Line 89: Modify to '… 2) perform a sensitivity analysis on the new model'.

*Response to comment*: We have revised it as the reviewer suggested.

Line 90: Modify to '… in modeling hydrological process in seasonally frozen soils.'

*Response to comment*: We have revised it as the reviewer suggested.

Material and Methods:

Line 93: Should be '… in northern China.'

*Response to comment*: We have revised it as the reviewer suggested.

Line 95: Should be 'Field experiments at…'.

*Response to comment*: We have revised it as the reviewer suggested.

Line 128: Should be 'During the soil freeze-thaw period at…'.

*Response to comment*: We have revised it as the reviewer suggested.

Line 192: you not describe what the pF value is.

*Response to comment*: We have revised it as "pF value (i.e. the logarithm of the absolute value of soil matric potential)".

Line 229: Change to 'latent heat transfer:'

*Response to comment*: We have revised it as the reviewer suggested.

Line 257: Lateral boundaries? Isn't Coup a 1-D model? Why would it need a lateral boundary condition?

*Response to comment*: We have deleted "and lateral".

Line 285: You refer to equation (5), but are talking about the surface energy balance, I think you meant equation (15).

*Response to comment*: Yes, we meant to equation 15 and we have revised it.

Line 291: need a citation for the Richardson equation as readers may not be familiar with the relationship.

*Response to comment*: We have added the following reference to the Richardson equation.

Richardson, H., Basu, S., and Holtslag, A. A. M.: Improving stable boundary-layer height estimation using a stability-dependent critical bulk Richardson number, Bound.-Lay. Meteorol., 148, 93–109, doi:10.1007/s10546-013-9812-3, 2013

Line 323: I think you mean H is the total sensible heat stored in the soil, not total energy?

*Response to comment*: Yes, H is sensible heat flux and calculated from the total heat storage E.

Results and Discussion:

Figure 3: Figure 3a should be modified to use solid circles like Figure 3b. It would make the Fig. 3a easier to read.

*Response to comment*: We have revised it as the reviewer suggested.

Conclusions:

Line 543: Should be '… are coupled in agricultural fields'.

*Response to comment*: We have deleted this sentence.

Line 566: Replace 'would be very necessary in investigation' to 'is still needed to improve understanding of

*Response to comment*: We have revised it as the reviewer suggested.